# Discriminant Canonical Analysis of the Contribution of Spanish and Arabian Purebred Horses to the Genetic Diversity and Population Structure of Hispano-Arabian Horses

**DOI:** 10.3390/ani11020269

**Published:** 2021-01-21

**Authors:** Carmen Marín Navas, Juan Vicente Delgado Bermejo, Amy Katherine McLean, José Manuel León Jurado, Antonio Rodriguez de la Borbolla y Ruiberriz de Torres, Francisco Javier Navas González

**Affiliations:** 1Department of Genetics, Faculty of Veterinary Sciences, University of Córdoba, 14071 Córdoba, Spain; carmen95_mn@hotmail.com (C.M.N.); juanviagr218@gmail.com (J.V.D.B.); 2Department of Animal Science, University of California Davis, Davis, CA 95617, USA; acmclean@ucdavis.edu; 3Centro Agropecuario Provincial de Córdoba, Diputación Provincial de Córdoba, 14071 Córdoba, Spain; jomalejur@yahoo.es; 4Unión Española de Ganaderos de Pura Raza Hispano-Árabe, 41001 Sevilla, Spain; uegha@caballohispanoarabe.com

**Keywords:** canonical discriminant analysis, Wright’s F statistics, Nei genetic distances, horse, inbreeding, Arabian, Spanish, Pedigree structure analysis

## Abstract

**Simple Summary:**

The demographic and genetic diversity structure and the contributions of Spanish (PRE) and Arabian Purebred (PRá) horses to the process of conformation of the Hispano-Arabian (Há) horse breed were evaluated. Genetic diversity parameters (inbreeding coefficient, genetic conservation index, coancestry coefficient, non-random mating degree, relatedness coefficient, maximum, complete, and equivalent generations, and number of offspring) were evaluated using a discriminant canonical analysis to determine the partial contributions of each founder breed to the development of Há horse breed. The calculation of Nei genetic distances suggests the overlapping could be estimated in 29.55% of the gene pool of the Há having a PRE origin while 70.45% of the gene pool of the breed may derive from a PRá origin. Although a progressive loss of founder representation may have occurred, breeding strategies implemented considering mating between animals with the highest genetic conservation indices (GCI) may compensate for its effects.

**Abstract:**

Genetic diversity and population structure were analyzed using the historical and current pedigree information of the Arabian (PRá), Spanish Purebred (PRE), and Hispano-Arabian (Há) horse breeds. Genetic diversity parameters were computed and a canonical discriminant analysis was used to determine the contributions of ancestor breeds to the genetic diversity of the Há horse. Pedigree records were available for 207,100 animals born between 1884 and 2019. Nei’s distances and the equivalent subpopulations number indicated the existence of a highly structured, integrated population for the Há breed, which is more closely genetically related to PRá than PRE horses. An increase in the length of the generation interval might be an effective solution to reduce the increase in inbreeding found in the studied breeds (8.44%, 8.50%, and 2.89%, for PRá, PRE, and Há, respectively). Wright’s fixation statistics indicated slight interherd inbreeding. Pedigree completeness suggested genetic parameters were highly reliable. High GCI levels found for number of founders and non-founders and their relationship to the evolution of inbreeding permit controlling potential deleterious negative effects from excessively frequent mating between interrelated individuals. For instance, the use of individuals presenting high GCI may balance founders’ gene contributions and consequently preserve genetic diversity levels (current genetic diversity loss in PRá, PRE, and Há is 6%, 7%, and 4%, respectively).

## 1. Introduction

The prospects of complementing already established endangered breeds or populations into a multi-breed composite population was proposed by Shrestha [1]. This author argued that, even though the development of a composite population does not directly lead to the conservation of an endangered breed, it may promote ‘conservation by utilisation’, through the preservation of the inherent potential of foundational breeds.

The aforementioned concept lays the basis for breeding programs either it is conservation plans seeking the sustainable management of native breeds, or the commercial development and utilization of specific crosses for practical purposes. Particular features of each population contribute to the genetic variability of the species to which it belongs. However, these particular features could dilute due to sub structuring, intermixing and/or consequent genetic drift over time [2].

The origins of the Há horse breed [3] date back to the year 742, and can be set in the cavalry of Balŷ ibn Bišr al-Qušayri, eighteenth governor (valí) of al-Ándalus (741–742), who brought Arabian horses along with his 10,000 soldiers to the Iberian peninsula when Berber insurrection extended to al-Ándalus [4]. As a result of this introduction, even though a small number of pure Arabian horses (PRá) were brought by the army of Balŷ, they were sufficient to originate a new type of horse: the Iberian Arabian horse. These PRá horses from Spain crossed with the primitive Spanish horse, found in the peninsula, originated the predecessors of the current Há horse [4].

Since its origins, Há horse breeders have sought complementarity between PRE and PRá horses [5]. As suggested by some authors [5], PRE horses genetically provided an optimal endocrine functionality and regulatory ability of the muscular physiology, together with an enhanced reproductive efficiency and adaptability to harsh environments, which promoted the expansion of the breed to the New World with Cristopher Columbus in 1493. Simultaneously, PRá horses conferred their strength, dexterity, and their cognitive superiority [6] to the crossing.

Both ancestor breeds have been reported to be more closely related to each other than other affine breeds such as Anglo-Arabian or The English Trotting Horse, as suggested by the genic and genotypic frequencies for blood systems “A” and “O” [7,8,9].

The improved features derived from composite vigour (heterosis) prompted authors such as Ibn al-’Awwam, agriculturist of the later 12th century, to refer to the cross between PRE and PRá as “the best” par excellence in his Kitāb al-filāha [10]. This manuscript is the most comprehensive agricultural treatise in Arabic and gathers all the knowledge of its time in respect to agriculture, horticulture and animal husbandry [11]. Contextually, some authors have hypothesized that Há may have been an attempt to emulate other relevant Arabian breeds of the Middle East [12].

At first sight, Há horses may have a smaller size, a remarkable trend to fit eumetric proportions and rather lively gaits than its ancestor breeds (PRE and PRá horses). There are some archive records reporting the use of earlier Há breeding studs, as far back as 1778. However, the formal creation of the breed dates to the 1883, when an official breeding program was implemented coinciding with the introduction of Arabian horses to upgrade and expand other horse breeding programs. Although the breed is linked to the Andalusian countryside, where it gained popularity as a working cattle horse, its expansion and consolidation as a breed has been slow [13]. The lack of quality of stock in some of the earlier breeding lines and the popularity of the cross between Há and thoroughbred (part-bred) horses to conform the Tres Sangres (Three bloods) composite breed can be found among the main causes for this slow consolidation process.

The Há breed was formerly very common in the south of Spain (Andalusian), area in which it was presumably created and from which the breed would spread even internationally. However, an increased national and international demand for the PRE horses led to the decrease in Há breed effective numbers, until it became a minority rare breed by the mid-1980s. This situation led to the constitution of the Há horse breed studbook in 1986 and its later official recognition within the category of special protection through the inclusion of the breed in the Official Catalogue of Spanish Cattle Breeds (Orden APA/2129/2008) [14]. The efforts of Cría Caballar (Military stables), a division of the Spanish Ministry of Defence, started in 1990 with a number of strategies which aimed to conserve and aid the recovery and breeding of the Há breed on military stud centers. As a result, in the middle of the 19th century the breed started to be considered standardized, due to the role it played in military campaigns. The current breed standard was published in 2002, and was modified in 2005. Since 2008, the studbook has been held by the Spanish Union of Purebred Hispano-Arabian Horses Breeders (UEGHá).

Contextually, the physical endurance, athletic qualities, balance in character of the PRá horse were, have been and still are combined with the versatility, predisposition to work and movement precision of the PRE horse. The evolution of the breeding strategies sought the recovery of the numbers of Há horses and can be evaluated in five stages. The first stage ranged from 1992 to 1995 and was marked by the use of 16 foundation Há stallions bred with (F1) 50:50 mares, resulting from the crossing between PRE dams and PRá sires. Afterwards, during the second stage which ran from 1996 to 1999, five selected PRá stallions were used to cover Há mares and one PRE stallion was selected to cover PRá mares. The third stage (2000–2005) was characterized by the predominant use of PRE horses covering more than 80% of the Há mares. A parallel breeding program dedicated to producing F1 Há individuals from PRE mares covered by both PRá stallions and five Há stallions of 50% and 75% PRá blood ratios obtained at the aforementioned stages. Simultaneously, the military covered PRá mares with PRE stallions, which had not been a frequent practice at the previous stages. Additionally, alongside these controlled breeding strategies, some mares were freely covered by Há stallions to produce F2 and F3 individuals optimally fitting the breed standard with the aim to improve the breed.

The objective of the fourth stage, which run from 2006 until our days, was to include the F2 individuals with PRá blood percentages ranging between 62.5% and 37.5%. Such individuals used to be difficult to find at previous stages, but are the natural progression of outbreeding 25% and 75% Há horses to PRE or PRá horses. Contextually, Há stallions and mares of 25% and 75% PRá blood bred to each other produce a 50:50 fixed Há genotype. This strategy sought to ensure that the products derived from the aforementioned breeding program effectively ascribed to the breed standard. At the age of 3, individuals start to be trained in the development of the skills comprised by the disciplines of Spanish cowboy/western dressage (Doma vaquera) and cattle driving, for which it stands out above other breeds (such as its ancestor breeds, PRE and PRá horses) [12].

Although the evaluation of genetic diversity and demographic parameters raised research interest in the past [15], previous analyses did not consider the repercussions of each of the two ancestor breed. Hence, parameters may have potentially been misestimated as a result of the genealogical information present in pedigrees being largely incomplete [16,17].

Therefore, the aim of this study is the development of a model to perform the discriminant canonical analysis of the contributions of PRE and PRá horses to the historical and present genetic diversity and demographic structure of the Há horse breed. Pedigree completeness was evaluated downwards, checking the repercussions of ancestors and founders, evaluating the structure of the population, its genetic variability, and connections between its genetic and demographic parameters, measuring the existing gene flow, and quantifying the risk of genetic diversity loss. The endangerment risk that the Há horse breed faces was evaluated to suggest effective conservation strategies, which in the case of the Há composite breed may reinforce its entity within the horse breed international panorama.

## 2. Materials and Methods 

### 2.1. Data Registries and Software Tool

Since the Hispano-Arabian (Há) horse breed is the product that results from the cross between the Spanish (PRE) and Arabian (PRá) purebred horses, the historical pedigree files for the three breeds were considered to build the historical pedigree database used in this study. The historical dataset comprised a total of 207,100 horses. Há historical pedigree database file was supplied by the Spanish Union of Purebred Hispano-Arabian Horses Breeders (UEGHá) and comprised 11,010 individuals, 4268 males and 6742 females, born between January 1950 and April 2019. The PRE historical pedigree database was supplied by the National Association of Purebred Spanish Horse Breeders (ANCCE) and comprised 172,797 individuals—83,408 males and 89,389 females, born between January 1884 and July 2019. Spanish PRá historical pedigree record was provided by the Spanish Association of Arabian Horse Breeders (AECCA). PRá historical pedigree database comprised a total of 23,293 individuals—11,143 males and 12,150 females, born between January 1898 and June 2019.

All the analyses were performed on a population set from which all death animals had been removed to evaluate the evolution and trends described by diversity and population structure parameters. This population set comprised the currently living populations of the three breeds (Há, PRE, and PRá). Há current pedigree database file comprised 9997 individuals—4031 males and 5966 females, born between December 1984 and April 2019. PRE current pedigree comprised 141,357 individuals—69,184 males and 72,163 females, born between April 1984 and July 2019. Spanish PRá current pedigree record comprised a total of 13,576 individuals—6632 males and 6944 females, born between June 1985 and June 2019. A full description of the composition of the historical and current datasets is presented in Table 1 and Table 2. All the analyses were performed using ENDOG (v4.8) software [18] on the historical pedigree (both living and death individuals) and on the currently living population.

### 2.2. Genealogical Information Evolution

We studied the number of births to compute the maximum and mean number of offspring per stallion or mare. The pedigree completeness index (PCI) was computed summarizing the proportion of each ascending generation’s known ancestors, through the maximum number of traced generations; the number of complete traced generations; the number of complete equivalent generations, calculated as (1/2*n*) where *n* is the number of generations setting the individual apart from each known ancestor [19], equal to $\sum_{a=1}^{n_{b}}\frac{1}{2^{g_{ab}}}$ where *n_b_* is the total number of ancestors of the animal, *b* and *g_ab_* is the number of generations between *b* and its ancestor *a* [20]; and the pedigree information quality assessing the proportion of pedigree registered known parents, grandparents, great-grandparents, and great-great-grandparents. Generation intervals [21] and the mean age of parents when their offspring were born were calculated for the 4 gametic pathways: stallion to colt and stallion to filly and mare to colt and mare to filly, from every animal’s date of birth registry together with those of its parents’. The stallion/mare ratio was calculated considering the percentage of mares and stallions with breeding progeny and the number of breeding animals selected.

### 2.3. Inbreeding, Coancestry, and Assortative Mating Degree

Individual inbreeding (F) was computed using the methods in Meuwissen and Luo [22]. Each individual’s average relatedness (AR) was calculated as Gutiérrez et al. [18]. According to Leroy et al. [23], F and coancestry (C) coefficients are identity estimators by descent (IBD), a probability that differs whether the alleles considered belong to a single individual or two individuals, respectively. The individual rate of inbreeding ($\bar{\Delta F}$) for the generation, is calculated according to Gutiérrez et al. [24] through $\Delta F_{b}=1-\;\sqrt[{t_{b}-1}]{1-F_{b}}$, where *t_b_* is the number of complete equivalent generations and *F_b_* is the inbreeding coefficient of the individual b. The individual rate of coancestry ($\bar{\Delta C}$) for the generation was computed following the 
methods in Cervantes et al. [25] through $C_{ba}=1-\sqrt[{\frac{tb+ta}{2}}]{1-C_{ba}}$, where *t_b_* and *t_a_* are the number of equivalent complete generations and *C_ba_* is the coancestry coefficient for the individuals b and a. The degree of assortative mating (non-random mating of individuals having more genetic or phenotypic traits in common than likely in random or disassortative mating), was computed following the methods of Caballero and Toro [26], through (1−F)=(1−C)(1−α) [27,28].

### 2.4. Ancestral Contributions and Probabilities of Gene Origin

The effective number of founders (*f_e_*), was calculated using $f_{e}=\frac{1}{\sum_{k=1}^{f}q_{k}^{2}}$, where *q_k_* is the probability of gene origin of the *k*^th^ founder and *f* is the real number of founders [29]. The effective number of ancestors (*f_a_*), was determined by $f_{a}=\frac{1}{\sum_{k=1}^{f}p_{k}^{2}}$ where *p_k_* is the marginal contribution of a *k*^th^ ancestor [20]. The effective number of founder genomes (*f_g_*) was computed as the inverse of twice the average coancestry as reported in Caballero and Toro [26]. The expected marginal contribution of each major ancestor *j* was computed as Boichard, et al. [20] and the contributions to inbreeding of nodal common ancestors (inbreeding loops), were computed according to Colleau and Sargolzaei [30]. The mean effective population size ($\bar{N_{e}}$) [27] was calculated as $\bar{N_{e}}=\frac{1}{2\bar{\Delta IBD}}$. The number of equivalent subpopulations was computed according to Cervantes, et al. [31] using $S=\frac{\bar{N_{e}C_{i}}}{N_{e}F_{i}}$, where $\bar{N_{e}C_{i}}=\frac{1}{\left(2\bar{\Delta C}\right)}$, is the mean effective population size considering the coancestry coefficient [32] and $\bar{N_{e}F_{i}}=\frac{1}{\left(2\bar{\Delta F}\right)}$, considering the inbreeding coefficient. Genetic diversity (GD) was calculated using $GD=1-\frac{1}{2f_{g}}$. GD lost in the population since the founder generation was estimated using 1−GD. GD loss derived from the unequal contribution of founders was estimated as Caballero and Toro [26] using $1-GD^{*}$, where $GD^{*}=1-\frac{1}{2f_{e}}$. The difference between *GD* and *GD** indicates the *GD* loss owed to genetic drift accumulated since the foundation of the population [29], and the effective number of non-founders (*N_ef_*) was computed using $N_{ef}=\frac{1}{\frac{1}{f_{ge}}-\frac{1}{f_{e}}}$ considering the formula proposed by Caballero and Toro [26]. CFC version 1.0. was used to perform the analysis of ancestral contributions and probabilities of gene origin [33].

### 2.5. Canonical Discriminant Analysis (CDA)

A canonical discriminant analysis (CDA) was performed on genetic diversity parameters derived from pedigree analyses of inbreeding (F, %), average relatedness (ΔR, %), average coancestry (C, %), non-random mating rate (α), genetic conservation index (GCI), number of maximum, complete and equivalent generations and number of offspring per individual using the breed (PRE, PRá, and Há horse) to which each animal belonged as a labeling classification criteria, to measure the variation of the genetic diversity parameters estimated, and to establish, identify and outline within population clusters [34,35,36,37]. Hence, we determined the percentage of correctly allocated individuals in their populations of origin in comparison to those animals which were statistically misclassified or attributed to a different breed from the one to which they belonged; to discover a linear combination of genetic diversity parameters that provide maximum separation between the potentially existing different groups when the classification criterion was the breed of the individual. CDA was also used to plot pairs of canonical variables to help visually interpret group differences. Variable selection was performed using regularized forward stepwise multinomial logistic regression algorithms.

The choice to perform a forward stepwise analysis was made considering the following alternatives. On the one hand, the first option considered was to perform a regularized canonical discriminant analysis. Regularization has been reported to improve the estimate of covariance matrices in situations where the number of predictors is larger than the number of data, as in such cases regularization may lead to an improvement of the efficiency of discriminant analysis. However, this was not our case as the nature of the variables considered may lead to the occurrence of considerable problems of multicollinearity. Such multicollinearity problems may derive from the fact that some of the variables initially considered, were computed including others (which were included too) among the terms in their formulas. As a result, even if models were simplified, removed variables may still be considered somehow.

Additionally, the analysis used in the present study must be robust in cases of highly unequal sample sizes. To approach such compromising situation, we used the approximation proposed by Roemisch et al. [38] of a regularized stepwise discriminant analysis. Unequal group sample sizes may affect the quality of classification, not axes. For these reasons, as suggested by Tai and Pan [39] and Roemisch, et al. [38] priors were regularized based on group sizes using the compute from group sizes from the prior probability option in SPSS version 25.0 [40] instead of considering them to be equal.

Furthermore, even if unequal sample sizes are acceptable as reported by Poulsen and French [41], some requirements must still be fulfilled. For instance, the sample size of the smallest group needs to exceed the number of predictor variables. As a “rule of thumb”, the smallest sample size should be at least 20 for each 4 or 5 predictors. The maximum number of independent variables is n−2, where n is the sample size. Although such a low sample size may be valid, it is not encouraged, and generally it is best to have 4 or 5 times as many observations and independent variables for discriminant approaches to be efficient. The present study satisfies this requirement by far, hence the potential distorting effects derived from unequal group sample sizes comparison are avoided.

On the other hand, after the stepwise canonical discriminant analysis approach was chosen, a decision had to be made on either to perform a backward or forward stepwise variable selection approach. As a drawback, stepwise residual sum of squares will typically be above that for best subset, if there is correlation between the predictors considered. For this reason, we performed a multicollinearity analysis and correlated variables exceeding minimally acceptable levels were discarded. Contextually, it must be considered that when regressors (variables) are independent, that is, they are not correlated, the variables chosen by either forwards or backwards stepwise selection methods will be exactly the same.

Conclusively, forward stepwise selection method was chosen given it is rather effective (less time-consuming) than backward selection methods and provided observations do not need to be strictly greater than the number of variables, which may make this approach valid for future research facing the same sample contexts.

Canonical Discriminant Analysis was performed using the Discriminant routine of the Classify package of the software SPSS version 25.0 [40] and the Discriminant Analysis *(DA)* routine of the Analysing Data package of XLSTAT Pearson Edition [42]. 

#### 2.5.1. Multicollinearity Preliminary Testing

Before running a discriminant canonical analysis (CDA), multicollinearity assumption should be tested for, to ensure redundancies in the variables considered do not affect the structure of the matrices or overinflate variance explanatory potential. The variance inflation factor (VIF) was computed and used as an indicator of multicollinearity. Computationally, it is defined as the reciprocal of tolerance: 1/(1 − R^2^). A recommended maximum VIF value of 5 [43] and even 4 [44] can be found in the literature. VIF was computed using the Linear routine of the Regression package of the software SPSS, version 25.0 [40].

#### 2.5.2. Canonical Correlation Dimension Determination

A canonical correlation analysis is a multivariate analysis of correlation. Canonical is the statistical term for analyzing variables which are latent (not directly observed), but which represent multiple variables (which can be directly observed). The maximum number of canonical correlations between two sets of variables is the number of variables in the smaller set. The first canonical correlation explains most of the relationship between sets [45]. Canonical correlations are interpreted as Pearson’s ρ. Hence, squared canonical correlation (Rc squared) [46] is the percent of variance in one set of variables explained by the other set along the dimension represented by the given canonical correlation (usually the first), that is the percent of shared variance along this dimension (analogous to R squared in multiple regression) [47]. As a rule of thumb, meaningful dimensions are detected when their canonical correlation are ≥0.30, which corresponds to about 10% of explained variance. All meaningful and interpretable canonical correlations should be reported, despite reporting of only the first dimension being common in research [48].

#### 2.5.3. Canonical Discriminant Analysis Efficiency

Wilks’ Lambda test assesses which variables significantly contribute to the discriminant function. As a rule of thumb, the closer Wilks’ lambda is to 0, the higher is the contribution of that variable to the discriminant function. Wilk’s Lambda significance can be tested using χ^2^. When significance is below 0.05, the corresponding function can be concluded to explain group adscription well [49].

#### 2.5.4. Canonical Discriminant Analysis Model Reliability

Pillai’s trace criterion was used to test the assumption of equal covariance matrices in discriminant function analysis (DFA). As smaller significance levels (*p* < 0.001) are considered [50] and sample sizes are unequal, Pillai’s trace criterion is the only presumably acceptable test to determine equality of covariance matrices [51]. Pillai’s criterion (as opposed to Wilk’s lambda) when used to test large samples, randomly deletes cases from the sample to equalize the numbers in each group, which enables assuming power can be maintained at a sensible level. Furthermore, Pillai’s Trace test is very robust and not highly linked to assumptions about the normality of the distribution of the data and is also preferable if we had violated the assumption of homogeneity of variance-covariance. Pillai’s criterion was computed using the Multivariate routine of the General Linear Model package of the software SPSS, version 25.0 [40]. In general, a significance below 0.05 means that there is a significant difference in the dependent variables (genetic parameters) across the levels of independent variables being tested, in our case the breed factor and its levels or possibilities (PRá, PRE, and Há horse breeds) [52].

#### 2.5.5. Variable Dimensionality Reduction

A preliminary principal component analysis (PCA) was performed to minimize overall variables into few meaningful variables that contributed most to variations in the breeds.

#### 2.5.6. Canonical Coefficients and Loading Interpretation and Spatial Representation

Discriminant function analysis was used to determine the percentage assignment of individuals into their own breeds. The traditional approach to interpreting discriminant functions examines the sign and magnitude of the standardized discriminant weight (also referred to as a discriminant coefficient) assigned to each variable in computing the discriminant functions. Small weights may indicate either that a certain variable is irrelevant in determining a relationship or that it has been discarded because of a high degree of multicollinearity with the rest of variables.

Discriminant loadings represent the variance shared between independent variables and the discriminant function. Discriminant loadings can be interpreted as factor loadings to evaluate the relative contribution of each independent variable to the discriminant function. Variables exhibiting a discriminant loading of ≥|0.40| are considered substantially discriminating variables. Stepwise procedures may prevent non-significant variables from entering the function. Simultaneously, multicollinearity and other factors may preclude a variable from entering the equation, which does not necessarily exclude that it has a substantial effect. Loadings are relatively more valid than weights to interpret the discriminating power of independent variables due to their correlational nature.

The comparison between variables measured on different scales can be performed considering standardized coefficients. Large absolute coefficients will denote a better discriminating ability. Discriminant scores can be computed by using the standardized discriminant function coefficients applied to data that have been centered and divided by the pooled within-cell standard deviations for the predictor variables, as discussed in IBM Corp. [53].

The data were standardized following the standard procedures described by Manly [54] before squared Mahalanobis distances and principal component analysis were calculated. Squared Mahalanobis distances were computed between populations using the following formula:$$D_{ij}^{2}=\left(\bar{Y_{i}}-\bar{Y_{j}}\right)\;COV^{-1}\left(\bar{Y_{i}}-\bar{Y_{j}}\right)$$
where $D_{ij}^{2}$ is the distance between population i and j and COV^−1^ the inverse of the covariance matrix of measured variable x and $\bar{Y_{i}}$ and $\bar{Y_{j}}$ are the means of variable x in the ith and jth populations, respectively. The Mahalanobis squared distance, defined as the square of the distance between centroids, was used to determine the existence of significant differences in the values for genetic diversity parameters across the three breeds [55]. Additionally, to confirm such differences, Nei’s minimum genetic distances [56] among the individuals of the breeds were computed. Dendrograms for PRá, PRE, and Há breeds were constructed using the construct Unweighted Pair-Group Method using Arithmetic averages (UPGMA) Tree task from the Phylogeny procedure of MEGA X 10.0.5.

#### 2.5.7. Discriminant Function Cross-Validation

The percentage of correctly classified cases is called the hit ratio [57]. To establish whether the percentage of correctly classified cases is enough as to consider discriminant functions issue valid results, as a form of significance, the leave-one-out cross-validation option was used. As reported by Schneider [58], the leave-one-out cross validation is K-fold cross validation taken to its logical extreme, with K equal to N, the number of data points in the set. That means that N separate times, the function approximator is trained on all the data except for one point and a prediction is made for that point. In this context, the classification rate of a cross validated discriminant analysis should be at least 25% greater than that obtained by chance for classification accuracy to be considered sufficiently achieved.

The validity of cross-validation can be supported by Press’s Q significance test of classification accuracy for original against predicted group memberships. In opposition to *t*-test for groups of equal size, in Press’ Q statistic, groups (breeds in our case) can be of unequal size [59]. Press’ Q statistic can be used to compare the discriminating power of a cross validated function to a model classifying individuals at random (50% of the cases correctly classified), as follows:Press’ Q = [N − (nK)]^2^/N(K − 1)(1)
where N is the number of individuals in the sample, n is the number of observations correctly classified, K is the number of groups. Afterwards, the value of Press’ Q statistic should be compared to the critical value of 6.63 for χ^2^ with one degree of freedom at a significance of 0.01. Under this assumption, when Press’ Q exceeds the critical value of χ^2^ = 6.63, cross-validated classification can be regarded as significantly better than chance.

## 3. Results

### 3.1. Genealogical Information Evolution

A progressively increasing trend was detected for birth number across PRá, PRE, and Há horses running from 1944 to 2006 when number of births in the three breeds dramatically decreased (Figure 1). The historical number of births was noticeably higher in PRE horses when compared to PRá horses and Há horses [60,61].

The number of complete generations for each of the three breeds (PRá, PRE, and Há) in the historical population was 3.35 ± 1.43, 4.59 ± 1.43, and 2.96 ± 1.42, respectively. However, for the currently living population the numbers of complete generations were 4.03 ± 1.42, 4.86 ± 1.42, and 3.04 ± 1.42, respectively. The number of equivalent generations in the three horse breeds was 5.76 ± 2.09, 8.36 ± 2.09, and 6.15 ± 2.08, respectively. For the currently living population, the number of equivalent generations were 7.11 ± 2.09, 8.85 ± 2.08, and 6.34 ± 2.08, respectively. As can be observed, the average number of equivalent generations, that is the number of generations separating the individual apart from each known ancestor [62], tends to converge in the historical and currently living populations. This situation may be promoted as the animals presenting rather incomplete pedigrees are also the oldest ones, and have died in most of the cases, hence they are no longer considered to compute currently living population’s parameters, which only comprises living individuals. However, the number of complete generations, or number of generations separating an individual from the furthest generation for which two of its ancestors are known [62], was around half the number of equivalent generations, which could have been expected, but which suggests that even if the genealogical information of individuals has progressively increased through the years, incomplete and partially incomplete pedigrees are still representative in the population from around the fourth to the fifth generation on, which can also be evidenced by the decrease in pedigree completeness occurring from the fourth generation on shown in Figure 2.

Pedigree completeness index (PCI) for the first generation (know parents) experienced a slight increase between 6.81–0.06% when the historical and current populations were compared. Minimum PCI (58.01%) was reached at the fifth generation (great-great-grandfather acquaintances) in the historical population of the Há horse breed. By contrast, the maximum PCI was reached at the first generation (known grandparents) in the current population of the Spanish horse breed (99.99%). A summary of the analysis of the maximum number of traced generations, PCI (first, second, third, fourth, and fifth generation), number of maximum generations, number of complete generations, and number of equivalent generations in the studied population sets is shown in Table 1.

Although Table 1 presents values for PCI for the Historical and currently living populations of PRá, PRE, and Há horses, separately, genetic diversity and structure parameters were computed considering all of the animals as a single population, given the ancestors of Há horses are comprised in PRá and PRE horse populations. To clarify this, Figure 2 depicts the evolution of pedigree completeness (PCI), inbreeding (F), coancestry (C), and population size (n) across maximum generations. 

Figure 2 shows that PCI describes the same trends in both historical and currently living population, with a slight increase in currently living values. This suggests unknown pedigree knowledge may be typical of ancestral individuals, and may stop being representative once those individuals disappear, as their descendants progressively increase pedigree knowledge after each generation. Furthermore, inbreeding and coancestry levels may be almost constantly maintained across generations until the 21st generation is reached, even in those cases in which generations comprised considerably higher numbers of individuals.

Summary of the statistics derived from pedigree analysis including maximum progeny per stallion and mare, mean age of stallions and mares in reproduction and foals of stallions and mares selected for breeding is shown in Table 2, while generation intervals (years) for the four gametic routes in the PRE, PRá, and Há horses are shown in Table 3. The mean age (years) of the parents at the birth of their offspring for the four gametic routes in the PRE, PRá, and Há breeds are shown in Appendix A. The historical maximum progeny per stallion was considerably higher in PRE and Há horses compared to PRá horses. However, when currently living population sets were compared, maximum progeny in PRá horses only slightly reduced from 215 to 140 compared to PRE and PRá horses for which respective reductions in maximum progeny per stallion from 1804 to 366 and 1659 to 219 were reported. Comparatively, maximum progeny per mare was relatively similar for the historical and currently living population sets of the three breeds studied (Table 2). The average progeny per stallion in the historical population of PRá, PRE, and Há horse breed was 8.17 ± 14.76, 13.67 ± 26.52, and 18 ± 103.01, respectively. For the currently living population, the average progeny per stallion was 7.08 ± 11.04, 11.77 ± 18.50, and 11.42 ± 20.51, respectively.

The average progeny per mare in the historical population was 3.54 ± 3.10, 4.28 ± 3.40, and 2.73 ± 2.44 for PRá, PRE, and Há horse breed, respectively. Similarly, for the currently living population of the same breeds, average progeny per mare was 3.13 ± 2.55, 3.68 ± 2.73, and 2.45 ± 2.21, respectively. The minimum average progeny per mare was that of the Há currently living population 2.45 ± 2.21, whereas the maximum average progeny per mare was reported for the historical 4.28 ± 3.40 PRE breed population. Contrastingly, the minimum average progeny per stallion was found for the currently living PRá population 7.08 ± 11.04, while a maximum of 13.67 ± 26.52 was reported for the historical population of the PRE horse.

Progeny of stallions selected for breeding in the current population for PRá, PRE, and Há horse breed were 97.92%, 99.93%, and 80.94%, respectively, while the same parameter referred to mares reported values of 98.85%, 49.74%, and 94.08%, respectively. The lowest percentage for this parameter was obtained for the current population of stallions of the Há horse breed (80.94%) and the historical population of Spanish mares (48.27%). The average age of the stallions in reproduction for the historical population of PRá, PRE, and Há horse breeds was 22.14 ± 4.99, 23.87 ± 4.99, and 18.21 ± 4.99, respectively, while for their currently living populations was 23.32 ± 5.81, 25.31 ± 5.82, and 17.66 ± 5.74, respectively.

In the case of the historical population of PRá, PRE, and Há, average age of the mares in reproduction was 23.01 ± 4.84, 23.73 ± 4.97, and 21.65 ± 4.80, respectively, while average age of mares in reproduction in the currently living PRá, PRE, and Há populations was 24.55 ± 5.09, 24.17 ± 5.09, and 20.50 ± 4.03, respectively. The shortest age difference between males and females was reported for the PRE horse population (1.14 higher in stallions). However, the greatest difference was reported for the Há population, for which males in reproduction were 2.84 years younger than mares in reproduction, as shown in Table 2.

### 3.2. Inbreeding, Coancestry/Kinship and Degree of Non-Random Mating

Table 4 presents the number of inbred and highly inbred animals. Animals presenting any level of inbreeding different from 0 were considered to be inbred. However, as suggested by Beuchat [63], even if the deleterious effects of inbreeding begin to become evident at an inbreeding level of around 5%, it is when inbreeding reaches 10% that there is significant loss of vitality in the offspring as well as an increase in the expression of deleterious recessive mutations. Hence, animals presenting values over 10% for inbreeding were considered to be highly inbred animals.

Historical inbreeding levels for PRá, PRE, and Há horse breeds were 6.79%; 8.42% and 2.85%, respectively, with these values increasing in the currently living populations to 8.44%, 8.50%, and 2.89%, respectively (Table 4). Inbred animals in the historical population of the PRá, PRE, and Há horse breeds were 71.96, 33.48, and 43.11%, respectively while, for the currently living population, inbred animals were 39.84, 24.94, 5.18%, for the same populations respectively. Highly inbred animals historically and currently represent a significant percentage from inbred animals. Highly inbred animals were 48.08, 26.04, and 40.34% in the historical population of the PRá, PRE% and Há horse breed and, 19.41, 4.60% and 30.55% of the PRá, PRE, and Há horse breed, respectively. Additionally, the average coancestry or kinship coefficient was 0.72%, 5.62% and 2.13% for the currently living PRá, PRE, and Há horse breed populations.

Non-random mating degree was 0.06, 0.03, and 0.01 for the historical population of PRá, PRE, and Há horse breeds, respectively, while for the currently living population of the same breeds, non-random mating degree was 0.08, 0.03, and 0.01, respectively (Table 4).

Average coancestry in the historical population of the PRá, PRE, and Há horse breeds was 0.6%, 5.57%, and 2.06%, respectively. Values of average coancestry in the currently living PRá, PRE, and Há horse breed populations were 0.72%, 5.62%, and 2.13%, respectively (Table 4). The genetic conservation index (GCI) in the historical population of the PRá, PRE, and Há horse breeds was 9.65 ± 4.71, 9.34 ± 1.85, and 9.11 ± 5.91, respectively. The lowest GCI result was reported for the Há horse breed currently living population 9.38 ± 5.88. However, the highest value was registered in the currently living PRá horse breed population 11.50 ± 3.76.

### 3.3. Ancestral Contributions and Probabilities of Gene Origin

A summary of the results for the analysis of probabilities of gene origin, ancestral contributions and loss of genetic diversity is shown in Table 5 and Appendix A, respectively. The total number of founders in the currently living population reached values of 131, 18, and 347, for the PRá, PRE, and Há horse breeds, respectively. 

The effective number of ancestors was 56, 379, and 876 for the historical PRá, PRE, and Há horse breed, respectively (Table 5). Comparatively, the effective number of ancestors in the current population of the aforementioned breeds was 958, 192, and 847, respectively. The number of founder equivalent genomes (*f_g_*) in the historical PRá, PRE, and Há horse breeds was 11.02, 7.90, and 7.61, respectively. Current *f_g_* was 7.45, 13.47, and 13.94 for PRá, PRE, and Há horse breeds, respectively.

Numbers of founder genome equivalents (*fg*) translated into higher than 93% levels of genetic diversity (lower than 7% levels of genetic diversity loss) were found as reported in Table 5, for historical and currently living populations for each of the three breeds. The highest genetic diversity levels were found for the Há horse breed (96%). However, the lowest levels were reached in PRE horses (93%). Genetic drift since founders was responsible for from 1% to 2% of genetic diversity loss in the historical and current populations of the three aforementioned breeds. Simultaneously, genetic diversity loss that could be attributed to bottlenecks, genetic drift and the unequal contribution of founders ranged from 2% to 7%, with PRE horses suffering the highest losses in genetic diversity, followed by PRá horses and Há horse breed, respectively.

Considering the marginal genetic contribution of a single ancestor (identification number 199092) explained 15.48–30.37% of the gene pool of the population of PRá horses. For PRE horses, the single ancestor with identification number 194 explained from 15.00% to 29.56%, respectively. For the Há horse breed, the single ancestor with identification number 200,862 explained from 10.92% to 21.75%, respectively.

The numbers of equivalent subpopulations reported in the currently living population for PRá, PRE, and Há horse breeds were 1.58, 0.18, and 0.44, respectively. The parameters referring to the effective size of the population calculated through the individual inbreeding rate and coancestry rate, as well as the number of equivalent subpopulations can be found in Appendix A. The effective population size calculated through the individual inbreeding rate in the current PRá, PRE, and Há horse breeds, was 43.8, 48.54, and 53.19, respectively. By contrast, the effective population sizes calculated through the individual coancestry rate of the currently living population for PRá, PRE, and Há horse breed were 69.44, 8.90, and 23.47, respectively. Contrastingly, the effective population size calculated through the individual inbreeding rate in the historical PRá, PRE, and Há horse breeds were 49.02, 49.02, and 52.63 respectively. By contrast, the effective population size calculated through the individual coancestry rate of the historical population for PRá, PRE, and Há horse breeds was 83.33, 8.98, and 24.27, respectively.

### 3.4. Genetic Relationships between Breeds

The average Nei genetic distance between PRá, PRE, and Há horse breed historical and current populations was 0.013 and 0.025, respectively (Figure 3). The mean historical and current coancestry within subpopulations, when the criteria for subdivision was the breed, was 0.06 and 0.05, respectively. Mean historical and current coancestry levels in the metapopulation when the breed was chosen as the population subdivision criteria were 0.05 and 0.02, respectively. A summary of parameters measuring interpopulation relationship is presented in Appendix A.

Wright’s F statistics (Appendix A), the inbreeding coefficient relative to the total population (F_IT_) was 0.03 for the historical population and 0.04 for the currently living population when breed was chosen as the subdivision criteria. The inbreeding coefficient relative to the subpopulation (F_IS_) varied from 0.02 for the historical population to 0.01 for the current population (Appendix A). The correlation between random gametes drawn from the subpopulation relative to the total population (F_ST_) was 0.01 for the historical population and 0.03 for the currently living population. 

### 3.5. Variable Dimensionality Reduction

No variable was discarded provided all component loadings for the genetic diversity parameters were above |0.5|. This suggested the fact that those variables presenting a potential redundant confounding nature had been previously discarded, for instance, when multicollinearity (VIF) was tested. 

### 3.6. Canonical Discriminant Analysis

#### 3.6.1. Canonical Discriminant Analysis Model Reliability

The value of *p* < 0.05 obtained for Pillai’s trace criterion suggests that there is a difference across all the three breeds considered in this analysis (Table 6). Afterwards, Wilk’s lambda statistic was used to assess whether canonical discriminating functions contributed significantly to the separation of treatments, that is, it was used to test the meaning of the discriminating function (Table 7).

Inbreeding (F, %), average relatedness (ΔR, %), average coancestry (C, %), non-random mating rate (α), genetic conservation index (GCI), number of maximum, complete and equivalent generations, and number of offspring per individual were included at a preliminary stage of the analysis performed in this study. Tolerance (1/R^2^) and variance inflation factor (VIF) were analyzed to identify those variables that were responsible for multicollinearity between variables. VIF estimation suggested the parameters of inbreeding (F, %) and equivalent generations (VIF > 4) should not be considered in the analyses. After the removal of these variance explanatory redundant variables, the results for tolerance and VIF can be observed in Table 8.

#### 3.6.2. Canonical Coefficients and Loading Interpretation and Spatial Representation

The canonical discriminant analysis identified two discriminating canonical functions. The first had a high discriminatory power, as denoted by the eigenvalue of 4.227. The results are presented in Table 9. The first function obtained explains 98.6% of total variance. The second function contributes to the explanation of variance with 5.6% of the information to the analysis, that is, very low.

The results for the tests of equality of group means to test for differences across breeds once redundant variables have been removed are shown in Table 10. The greater the value of F and the lower the value for Wilks’ Lambda, the better the discriminating power a certain variable has and the lower the rank position it presents. Those variables presenting equal values of lambda and F had equivalent discriminatory power, for instance, offspring number, average relatedness (AR), non-random mating degree (α) and number of maximum generations. Even if this happens, Table 8 and Table 11 suggest similarities may not derive from a multicollinearity problem, but may appear because the variables, indeed, have a similar discriminant power.

Once F and Wilks’ Lambda had been assessed, the magnitude of standardized coefficients was evaluated to determine whether there had been a reduction in the discriminant power of individual variables as a result of multicollinearity between pairs (Table 11). Multicollinearity implies a reduction in the separate discriminant power of each of the two variables involved in the multicollinear relationship.

As shown in Table 11, standardized coefficients for the variables genetic conservation index (GCI) and offspring number fell below 0.4, which evidenced a decrease in the discriminating power of the non-individual function due to genetic conservation index and offspring number being related, explaining a somehow redundant fraction of variability.

The greater the reduction in the standardized coefficient of a certain variable, the more important is the relationship between variables holding similar Wilks’ lambda and F values. In this context and as suggested by Hair Jr [59], absolute values below |0.3| are indicative of multicollinearity problems if F and Wilks’ Lambda have been previously evidenced to be similar for a certain pair of variables. Hence, the dissimilarities in F values and Wilk’s Lambda between GCI and offspring number suggested redundancies may not derive from multicollinearity problems. This is supported by the fact that when such variables were removed from the analysis, variance explicative power did not decrease but retained the same values. As a result, both variables were retained in the analyses. This decision was made basing upon the fact that the eigenvalue reported was considerably higher (4.227 and 3.982 if variables were included and excluded, respectively), hence, squared canonical correlations, and the percentages of shared variance were slightly higher (80.8% and 79.9% of shared variance for the first discriminant function (F1), which maximized the explanatory potential of the discriminant functions (Table 9).

Unstandardized coefficients, calculated on raw scores for each variable, are of most use when our aim is to cross-validate or replicate the results of a discriminant analysis or to assign previously unclassified subjects or elements to a group. The present analyses evaluate the potential misclassification of individuals belonging to previously defined populations as a way to define such populations themselves. Hence, standardized coefficients must be interpreted while unstandardized coefficients must be discarded [60]. Furthermore, the unstandardized coefficients cannot be used to compare variables or to determine which variables play the most relevant role in group discrimination, as the scaling for each of the discriminator variables (i.e., their means and standard deviations) usually differs. The maximum number of canonical discriminant functions generated is equal to the number of groups minus one. In the present study, the number of canonical discriminant functions was 2 for each series, as we used the three horse breeds as a labelling criterion. After the evaluation of standardized coefficients, the resulting discriminant functions were as follows:F1: 1.083 × AR + (−0.368) × GCI + (−0.257) × α + 0.475 × Number of Maximum Generations + (−0.380) × Number of Complete Generations + 0.008 × Offspring number
F2: 0.071 × AR + (0.227) × GCI + (0.513) × α + (−0.899) × Number of Maximum Generations + (0.887) × Number of Complete Generations + 0.062 × Offspring number

To determine which variable should be discarded out of each pair of variables for which a multicollinearity problem has been detected, discriminant loadings were evaluated. Discriminant loadings measure the existing linear correlation between each independent variable and the discriminant function, reflecting the variance that the independent variables share with the discriminant function. In this regard, discriminant loadings can be interpreted like factor loadings in assessing the relative contribution of each independent variable to the discriminant function. A graphical representation of discriminant loadings is shown in Figure 4, with those variables whose coloured area extends further apart from the origin being the most representatively discriminating ones.

A territorial map was created by plotting the discriminating values for each observation (Z) for the first function on the x axis and those values for the second discriminant function on the y axis. Figure 5 graphically depicts the canonical discriminant analysis of individuals across the three breeds. The overlapping between the two breeds is patent. However, Há horse breed population is closest to PRá horses as evidenced in Figure 3 and Figure 5. The calculation of Nei genetic distances suggests the overlapping could be estimated in 29.55% of the gene pool of the Há having a PRE origin while 70.45% of the gene pool of the breed may derive from a PRá origin.

Centroids describe the central observation for each breed group. The probability that an unknown case belongs to a particular group was calculated by measuring the relative distance of Mahalanobis to the centroid of a that particular population. To compute discriminant scores or centroids, the mean for each breed was substituted in the two first dimensions [61]. Then, to calculate the optimal cut-off point, that is, the probability of classification the procedures in Hair et al. were followed [52]. Then, appropriately classified cases were determined. Despite the overlapping appreciated, it could be observed that each of the centroids for the three breeds was remote from the rest, which may evidence the current differentiation among the breeds considered.

#### 3.6.3. Discriminant Function Cross-Validation

When classification and leave-one-out cross-validation matrices are evaluated, it can be observed that 89.54%, 98.77% and 49.95% of the horses had been correctly classified for PRá, PRE, and Há horse breeds, respectively. Cross-validation was performed and Press’ Q value was calculated as follows;
Press’ Q = [N − (nK)]2/N(K − 1) = 207,100 – (197,026 × 3)^2/207,100 (3 − 1) = 355,961.14.
where N = 207,100 is the number of individuals in the sample, n = 197,026 is the number of observations correctly classified, K = 3 is the number of groups (breeds).

Contextually, as Press’ Q statistic was above 6.63 (significance level of 0.01), χ^2^ critical value for one degree of freedom at a 95% confidence level (*p* < 0.01), predictions were significantly better than chance. Hence, there is a correct classification rate of at least 50% [64].

## 4. Discussion

The Há horse breed is a composite breed which derives from the cross between PRE and PRá horse individuals. To better understand Há horses, we may have to go back and study its ancestor breeds. This may enable the gathering of a very comprehensive amount of information to study the evolution of genetic diversity and of factors affecting it. The examination of the dendrogram constructed from Nei genetic distances and the spatial distribution reported after the discriminant analyses (Figure 3) suggest the existence of clear population structure, which may still be partially supported by its two ancestor breeds. In this context, a higher repercussion of PRá horses on Há and hence proximity between the breeds is evidenced when PRá is compared to PRE horses (Figure 3 and Figure 5).

The Mediterranean breeds considered in the study (PRá, PRE, and Há horse breeds) present strong differential physiological, behavioural and morphological differences [5,6,10,12,65] which are closely related to the geographical locations in which they locate and have developed along the course of History. This distinction and the closest link between PRá and Há horses was also supported at a genetic level by Pablo Gómez, et al. [66]. This close connection between North African horses and those inhabiting in the Southern territories of the Iberian Peninsula has been reported to have occurred since prehistorical times and before the commercial routes established by the Greek and the Phoenician colonies took place.

According to Aparicio Sánchez [67], such former connection was reinforced for centuries when, once commercial routes had been established, a significant horse exchange existed between North Africa and the Iberian Peninsula and extended at least from the arrival of the Romans in Iberia in 219/8 BC for the conquest of Hispania and up to the expulsion of the Moriscos, as decreed by King Philip III of Spain on 9 April 1609 [68]. In line with this historical context, even if the breeding practices seeking the obtention of Há horses only became more relevant from 1800 on, the present study evidences that the greater contribution of PRás to the development of Há breed and its genetic diversity may not only be patent from the very first crosses carried out after the first PRá horses came to the Iberian Peninsula in 742, but it may also continue in our times.

The drastic reduction in the number of PRE horses (Figure 1) was a direct consequence of the economic crisis whose effects in the equine sector became patent in 2008, when the real estate bubble burst [60]. According to Palomo [60], prior to 2008, PRE horses had been acquired as a luxury item by real estate developers or companies, who had no choice but to missell them when the economic crisis arose. Abandonment, giveaway prices or slaughterhouse became the destination of thousands of individuals, which also brought about the drastic stop of breeding practices. It was only from 2012 on that it began to rebound due to leisure or sport. At the worst of times, individuals were sold for 150 €, while the average price ranges between 3000 and 5000 €. Há horses did not suffer from the effects of the crisis as drastically as PRE horses as these were usually kept on large farms, sometimes as a secondary production alongside cow farming, hence they were sold more easily and at a better price given their versatility [61].

Geographical barriers or socio-political considerations may not have played an important role in the loss of diversity of the population given the low levels of genetic diversity loss found in our study, which could be ascribed to the process of genetic drift within the population. Overall, the generation interval in Há is rather long, with a global mean for the four selection paths of around 14.81 years. Nevertheless, this is in line with other studies considering the PRE horse breed (10.1 years) [69], largely coinciding with the values for PRE horses in our study, with generation intervals of 10.58 years in the current population. Likewise, the generation interval in PRá horse breed was 13.03 years, which was slightly higher than the values reported in literature [70]. This suggest that the unbalanced contribution of PRá horses to the development of Há horses, may also contribute to the fact that breeding practices and possibly reproductive physiology [71] may resemble those in the PRá horse more than those in the PRE breed, which may explain the slightly longed generation intervals.

PCI in the three breeds studied are very high for the first five generations (from parents to great-great grandparents), which is common to autochthonous horse breeds as supported by Giontella, et al. [72]. The high quality of the pedigree used in this study, provided its length and depth, enables an accurate calculation of genetic diversity parameter. The almost constant levels of inbreeding and coancestry reported until the 21st generation suggest that the analysis of genetic diversity and population structure analysis is accurate and valid and accurate conclusions can be drawn. As demonstrated by Duru [73], all parameters describing the probability of gene origin of a certain population are affected by pedigree depth. In fact, a suitable estimation of genetic variability widely depends on available and accessible pedigree information measured by pedigree completeness.

The use of robust pedigrees with completeness indices of around 80% have reported to allow reliable estimations of inbreeding values, resulting in medium to high correlations with genomic inbreeding in horse breeds [74]. Average values for PCI in PRá, PRE, and Há historical and current populations were 77.07%, 90.13%, 96.13%, 99.01%, 73.26%, and 74.71%, respectively. Todd, et al. [75] suggested that among the causes supporting the occurrence of these high correlations, a large proportion of the inbreeding coefficient in horse current populations may be accounted for by ancestors many generations back in the pedigree. Inbreeding to distant ancestors results in shorter runs of homozygosity regions which might not be captured unless very high densities of SNP are used. These authors [75] suggested that pedigree analyses can report inferences of inbreeding that are comparably accurate to those reported by SNPs analyses, when complete pedigrees large populations considerably exceed the number of genotyped individuals. Such event occurs as pedigree data may enable to make inferences for deceased individuals (such as the founders of the population), whose molecular DNA material cannot be recovered to perform genotyping. Furthermore, these complete pedigrees may allow to evaluate the trends described by genetic diversity alongside the history of specific populations, offering the opportunity to determine the effects of particular individuals over time on the fitness of their descendants.

The use of breeds with dissimilar performance characteristics enhances the opportunities for the maximization of their average genetic merit in the resulting product. This is taken advantage to meet requirements for specific functionality, production and/or marketing situations. When breeds used in the foundation of a composite breed do not contribute equally, as suggested by our results, since there is a loss of heterozygosity between the first and second generations [76]. Then, if inbreeding is avoided, further loss of heterozygosity in mated populations does not occur. This theoretical framework is well described by the populations evaluated in the present study. For instance, although mean inbreeding levels in the historical population of the Há and of its ancestor breeds, PRá and PRE horse breeds were 2.85%, 6.79%, and 8.42%, respectively, there has been a slight increase in inbreeding levels in the current populations of the same breeds to reach the levels of the 2.89%, 8.44%, and 8.55%, respectively.

Although genetic erosion may have occurred at a very low rate, the population of Há horses has remained stable. These results suggest, timely action must be taken to control the increase in inbreeding through the years in the two ancestor breeds as levels are starting to approach compromising levels, which may result in the expression of deleterious effects derived from inbreeding in the population. One of the most relevant effects resulting from reduced genetic diversity is inbreeding depression, which in turn, may end up compromising the performance of domestic animals [77]. Santana Jr, et al. [77] reported that the maximum level of inbreeding that could be absorbed by animals before detrimental effects begin to negatively affect performance is around 20%. However, deleterious effects related to diseases, reproduction or cognitive function may be patent when inbreeding levels are around 12.5% or above [78,79,80,81].

In this context, the large number of stallions used in reproduction may have contributed to the stabilization of inbreeding. Still, the presence of highly inbred animals in the population in percentages of 4.60%, 19.41%, and 30.55% in the currently living Há, PRE, and PRá horse populations implies the excessive use of certain individuals is still patent. These findings support the fact that currently, the major concern in managing the genetic diversity of the Há horse breed is the short-term decrease in genetic variability due to the loss of genetic contributions from founders and ancestors, more than the long-term effect of inbreeding itself as it was suggested in literature [82,83,84,85,86,87,88].

When aiming to develop a genetically healthy breeding strategy, ΔR is a very useful parameter since it allows breeders to preserve the genetic pool of a certain population [72,89]. Because of that, if the stallions and mares presenting the lowest levels of ΔR are mated (in our case PRá with ΔR 1.42% and Há horse breed with ΔR 4.25%), the inbreeding levels in their future offspring may be reduced, which may act balancing the gene contributions of the founders in the population, and consequently the genetic diversity.

Levels of genetic diversity in the present population are over 93% for the three breeds. However, in the context of Há horses as a composite breed whose studbook is still open, these high levels may not ensure the preservation of the gene pool of the founding population, as this population may constantly be changing [90]. In such cases, the genetic conservation Index (GCI) can help to determine the contribution of founders of each breed to the composite population. McManus, et al. [91] described that GCI computes the genetic contributions of all the identified founders; for this reason, it has been assumed that the animals, which, get higher values of GCI, also gather wider fractions of the gene pool of the founding population. In our case, the higher GCI value was for PRá horse breed (11.50 ± 3.76), (9.74 ± 1.28 for Spanish horse breed) and (9.38 ± 5.88 for Há horse breed), which supports the higher contribution to the founding gene pool of PRá horses than PRE horses to the conformation and development of the Há horse breed, as it has been suggested by our results and other authors [66].

The determination of Mendelian sampling of the non-founders of a population has recently been reported to be of help when determining partial inbreeding in equine populations [92]. Inbreeding can be broken down into the sources of the coancestry between the parents of each individual, which becomes even more relevant in the case of composite breeds whose ancestors may belong to differentiated breeds. In our study, the number of non- founder (*N_ef_*) for PRá, PRE, and Há horse breed were 10, 11.40, and 23.57, respectively. These values support those reported in the literature [26,93], even more so when these values are evaluated in the context of the inbreeding levels presented for the three horse breeds discussed. In line with these results, we may determine that relative founder contributions may tend to stabilize after a short number of generations, which has been described in Thoroughbred horses [94]. Either the population is closed or remains open, as is the case of the Há horse breed [17,92].

FAO/UNEP [95] proposed the general rule of maintaining rate of inbreeding per generation should not exceed 1–3% [96]. Higher rates fix deleterious recessive genes too rapidly for selection to eliminate them, and the vigour and fertility of the populations decrease. When inbreeding rate is below 1% populations have been partially purged of deleterious genes and tolerate higher rates of inbreeding, hence, animal breeders can safely ignore some inbreeding and random loss of genes. In endangered populations conservationists develop a rather conservative approach. The rate of loss per generation of heterozygosity due to inbreeding as measured by F is equal to 1/(2 *N_e_*), where *N_e_* is the effective population size.

The definition of *N_e_* is complex, but certain criterion must be considered to permit a correct interpretation of this parameter. For instance, the sex ratio must be equal and individuals must randomly mate. A number of additional “ideal” characteristics could be stated. In practice, *N_e_* is always smaller than the actual number of breeding individuals. Thus, *N_e_* must equal at least 50 if our aim is to keep inbreeding rate below 1%. Still, even if inbreeding rate is 1%, the loss of genetic diversity is appreciable after a few generations, and a gradual erosion of genetic variation cannot be avoided. Eventually, the population will become virtually homozygous, the time depending on *N_e_*. Consequently, 1% criterion must be viewed as short-term criterion. A population with an effective size of 50 will lose about a quarter of its genetic diversity after 20–30 generations, and along with this, much of its capacity to adapt to changing conditions [97].

To maintain a genetically healthy population in these situations, *N_e_* must be increased. FAO/UNEP [95] suggests that G must approximately equal to *N_e_*, G being the number of generations the population is likely to retain its fitness at a relatively high level. Still, to conserve short-term fitness, or to maintain short-term fitness in captive populations other criteria must be accounted for, given effective population size is considerably affected by unbalanced sex ratios, population size evolution, by a non-random distribution of progeny among families, and other characteristics of the breeding systems implemented.

In line with these suggestions, Leroy, et al. [98] have reported that restricting the number of generations when calculating effective size may be a good option for population monitoring, due to its effectivity to detect short-term changes in genetic diversity while it permits a generation scaled increased accuracy of the estimation of effective size, while reducing the bias related to ancestral pathways disequilibrium in pedigrees. Still, the consideration of very limited number of generations may not completely account for the lack of independence of family sizes across generations. As a result, for breeds with relatively complete pedigree records, as the ones in our study, the estimation of effective size via coancestry rate may be of interest to provide an evaluation of long-term changes in genetic diversity over long periods. For instance, the reduced numbers for effective size calculated via individual coancestry rate found in PRE, can be linked to the recent sharp bottleneck occurring in 2008 as a direct consequence of the economic crisis (Figure 1), as afterwards, there may have been an overrepresentation of popular individuals, which drastically increased individual increase in inbreeding rate and whose effects may still be present in the current PRE population.

As suggested by Gutiérrez et al. [99], valuable information concerning population structure can be inferred from the comparison between individual increase in inbreeding and coancestry rate pathways to compute effective population sizes. This occurs as the two parameters are assumed to be measures of the same accumulated drift process, from the founding population to the present time. For instance, as reported by Malhado, et al. [100] both measurements of effective population size would be asymptotically equivalent in an idealized population and the disagreement between them may be mainly caused by their differential ability to assess the effect of preferential mating.

The imbalance between effective population size calculated via individual inbreeding and via coancestry rate suggests that the introduction of new reproductive individuals and the design of a plan for recommendable breeding pairs within the breeds populations could be a viable alternative to maintain the lowest possible ΔR coefficients, thereby preventing the increase in the probability of inbreeding deleterious effects from occurring and increasing the future genetic variability and effective size of the breeds.

Contextually, Alderson [82] would suggest that, apart from controlling inbreeding to develop genetic management program in animal populations, the consideration of the GCI in breeding programs focuses and seeks to maximize the retention of the allelic richness present in the base population, but it does not take into account pedigree bottlenecks which may have occurred through non-founder individuals. In these regards, Sørensen et al. [101] demonstrated that the comparison of *f_e_* and *f_a_* could be used to assess the occurrence of changes in genetic drift and recent bottlenecks in a population, which are corroborated if (*f_e_/f_a_*)>1, respectively. In our study, *f_e_* and *f_a_* were 2.35, 1.54, and 1.48, respectively, for each of the horse breeds considered (PRá, PRE, and Há). This finding is indicative of the fact that genetic drift may not have been stable in the three horse breeds studied, with a progressive loss of founder representation, which had previously been reported in the literature [93,101]. Despite these values for *f_e_*/f_a_, our study confirms that an increasing trend has been described by GCI over time. Hence, the fact that founder representation is being progressively gained, not only in the Há horse breed, but also in its two ancestor breeds, may derive from the attempts of breeders and breeding associations to plan matings, trying to compensate for the aforementioned loss of founder representation.

## 5. Conclusions

The Há horse breed population has a defined genetic structure, which may still be partially supported by its two ancestor breeds with a higher representation of PRá horses. The unbalanced contribution of PRá horses to the development of Há horses may also contribute to the fact that breeding practices and possibly reproductive physiology adapts more to those carried in the PRá horse than the PRE breed, which may additionally explain the slightly lengthened generation intervals. Although genetic erosion may have occurred at a very low rate, the population of Há horses has remained stable. The major concern in managing the genetic diversity of the Há horse breed is the short-term decrease in genetic variability due to the loss of genetic contributions from founders and ancestors, more than the long-term effect of inbreeding itself. The large number of stallions used in reproduction may have contributed to the stabilization of inbreeding. Reduced ΔR may act balancing the gene contributions of the founders in the population, and consequently itsgenetic diversity. Although a progressive loss of founder representation may have occurred, breeding strategies implemented considering mating between animals with the highest levels of GCI may compensate for the aforementioned loss of founder representation.

## Figures and Tables

**Figure 1 animals-11-00269-f001:**
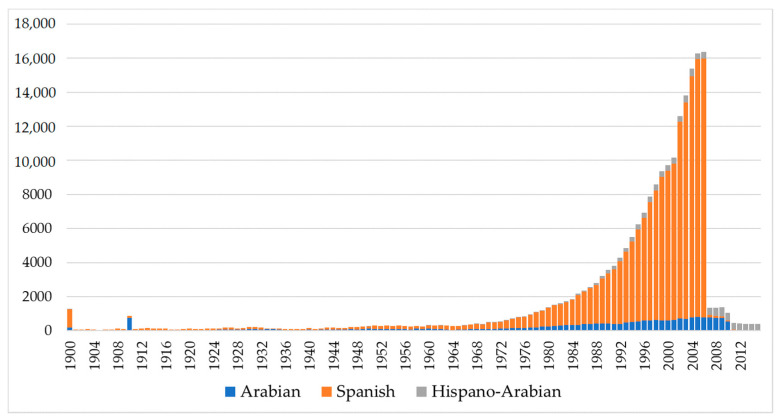
Historical birth number evolution in Hispano-Arabian (Há), Spanish (PRE) and Arabian (PRá) Purebred Horses from 1900 to 2019.

**Figure 2 animals-11-00269-f002:**
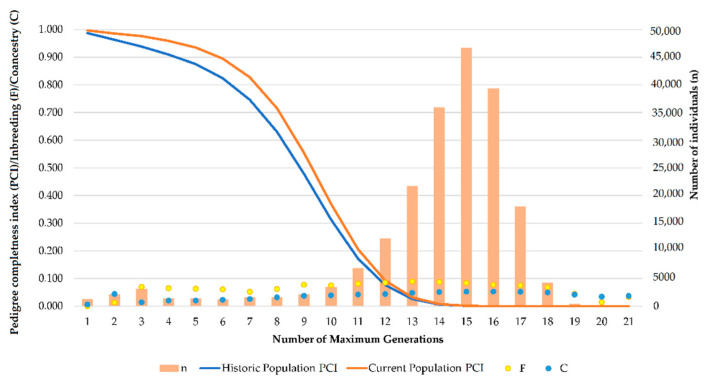
Evolution of pedigree completeness index (PCI), inbreeding (F), coancestry (C) and population size (n) in Spanish (PRE) and Arabian (PRá) Purebred and Hispano-Arabian (Há) horses until the 21st generation.

**Figure 3 animals-11-00269-f003:**
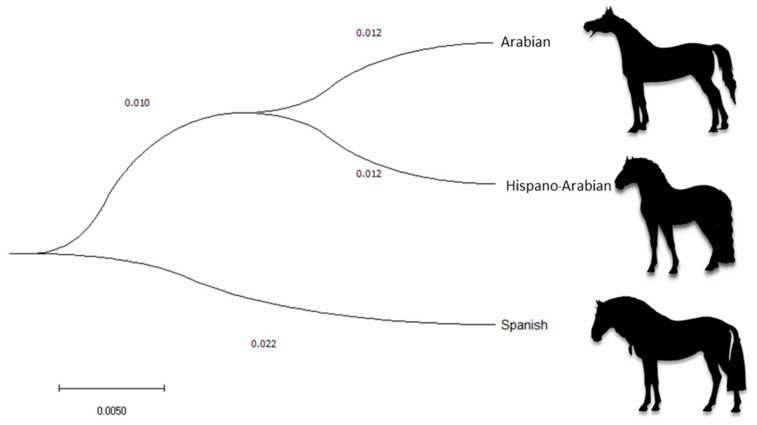
Cladogram constructed from Nei’s Distances between Arabian purebred (PRá), Spanish purebred (PRE), and Hispano-Arabian (Há) horse breeds.

**Figure 4 animals-11-00269-f004:**
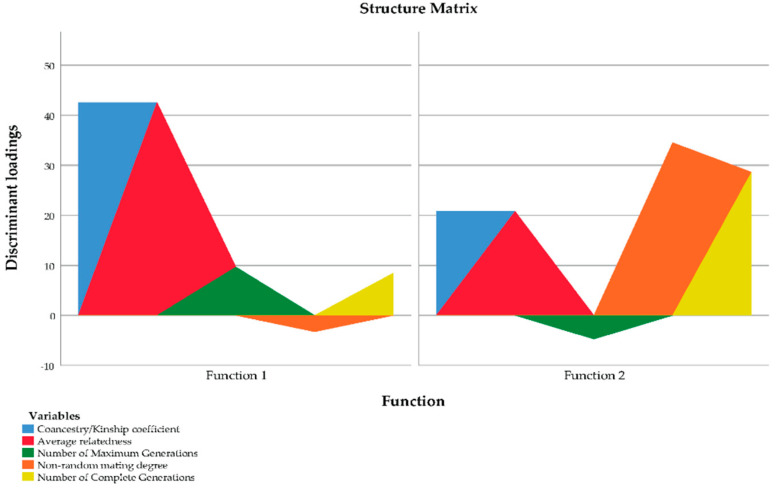
Vector plot of discriminant loadings for genetic diversity parameters.

**Figure 5 animals-11-00269-f005:**
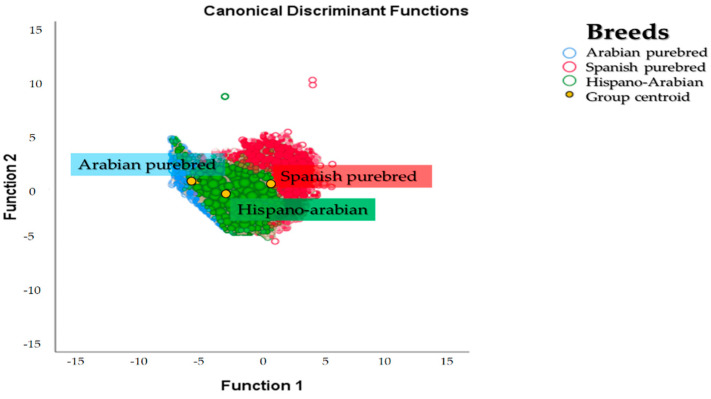
Territorial map depicting the results of the canonical discriminant analysis on the individuals comprising the Arabian purebred (PRá), Spanish purebred (PRE) and Hispano-Arabian (Há) horse breeds.

**Table 1 animals-11-00269-t001:** Statistics summary of the analysis of the pedigree, maximum number of traced generations, pedigree completeness (1st, 2nd, 3rd, 4th and 5th generation), number of maximum generations, number of complete generations and number of equivalent generations in the studied population sets for Arabian Purebred (PRá), Spanish Purebred (PRE) and Hispano-Arabian (Há) horse breeds.

	Population Set	Historical (n = 207,100)			Current (n = 164,941)		
Parameter		PRá	PRE	Há	PRá	PRE	Há
Population size		23,293	172,797	11,010	13,576	141,357	9997
Maximum number of traced generations, n		18	20	21	18	20	21
Pedigree completeness level at 1st generation, (Known parents)		92.08	99.77	96.4	98.89	99.99	96.46
Pedigree completeness level at 2nd generation, (Known grandparents)		85.33	98.61	83.71	94.70	99.86	85.26
Pedigree completeness level at 3rd generation, (Known great grandparents)		78.51	95.77	76.34	90.38	99.65	78.84
Pedigree completeness level at 4th generation, (Known great great grandparents)		71.07	94.99	64.13	85.21	98.80	66.38
Pedigree completeness level at 5th generation, (Known great great great grandparents)		64.57	91.93	58.01	83.29	96.81	58.55
Number of maximum generations (mean ± SD)		10.27 ± 4.89	14.28 ± 2.65	12.15 ± 5.75	12.86 ± 3.14	15.08 ± 1.38	12.54 ± 5.54
Number of complete generations (mean ± SD)		3.35 ± 1.89	4.59 ± 1.20	2.96 ± 1.78	4.03 ± 1.68	4.86 ± 0.95	3.04 ± 1.78
Number of equivalent generations (mean ± SD)		5.76 ± 2.78	8.36 ± 1.61	6.15 ± 3.04	7.11 ± 1.99	8.85 ± 0.83	6.34 ± 2.96

**Table 2 animals-11-00269-t002:** Summary of the statistics derived from pedigree analysis including maximum progeny per stallion and mare, mean age of stallions and mares in reproduction and foals of stallions and mares selected for breeding in the historical (n = 207,100) and current (n = 164,941) Arabian Purebred (PRá), Spanish Purebred (PRE) and Hispano-Arabian (Há) horse breed populations.

	Populational Sets	Historical			Current		
Parameters		PRá	PRE	Há	PRá	PRE	Há
Males%		47.84	48.27	38.76	48.84	48.95	40.32
Mean number of foals per stallion, (mean ± SD)		8.18 ± 10.31	13.67 ± 10.38	17.99 ± 103.01	7.08 ± 11.04	11.77 ± 18.50	11.42 ± 20.51
Maximum foal number per stallion, n		215	1804	1659	140	366	219
Average age of stallions in reproduction, years (mean ± SD)		22.14 ± 4.99	23.87 ± 4.99	18.21 ± 4.99	23.32 ± 5.81	25.31 ± 5.82	17.66 ± 5.74
Females%		52.16	51.73	61.24	51.16	51.05	59.68
Mean number of foals per mare, (mean ± SD)		3.54 ± 3.10	4.28 ± 3.40	2.73 ± 2.44	3.13 ± 2.55	3.68 ± 2.73	2.45 ± 2.21
Maximum foal number per mare, n		20	24	17	18	16	17
Average age of mares in reproduction, years (mean ± SD)		23.01 ± 4.84	23.73 ± 4.97	21.65 ± 4.80	24.55 ± 5.09	24.17 ± 5.09	20.50 ± 4.03
Male/Female Ratio		0.92/1	0.96/1	0.93/1	0.96/1	0.63/1	0.68/1
Progeny from stallions selected for breeding, %		91.77	99.20	92.66	97.92	99.93	80.94
Progeny from mares selected for breeding, %		92.02	48.27	94.20	98.85	49.74	94.08

**Table 3 animals-11-00269-t003:** Generation intervals (years) for the four gametic routes in the Spanish Purebred (PRE), Arabian Purebred (PRá) and Hispano-Arabian (Há) breeds.

	Parameter	Gametic Route	Stallion to Colt	Mare to Colt	Stallion to Filly	Mare to Filly	Total
Population Set							
PRá		N	2524	2480	5606	5579	16,189
		Mean	13.03	12.94	13.14	12.23	12.78
		SD	12.58	14.69	13.13	13.12	13.30
PRE		N	12,681	12,619	40,297	40,217	105,814
		Mean	10.58	9.68	10.67	9.56	10.12
		SD	6.82	5.48	6.87	5.95	6.40
Há		N	275	274	1643	1643	3835
		Mean	14.81	14.44	27.94	28.43	26.24
		SD	16.88	17.14	23.96	24.49	23.80

SD: Standard deviation; SEM: Standard Error of the Mean.

**Table 4 animals-11-00269-t004:** Statistics of pedigree analysis: inbreeding (F), average individual increase in inbreeding (ΔF, %), maximum coefficient of inbreeding (%), inbred and highly inbred animals (%), average coancestry (C, %), average relatedness (ΔR, %), non-random mating rate (α), and genetic conservation index (GCI).

	Populational Sets	Historical (n = 207,100)			Current (n = 164,941)		
Parameters		PRá	PRE	Há	PRá	PRE	Há
Inbreeding (F, %)		6.79	8.42	2.85	8.44	8.50	2.89
Average individual increase in inbreeding (ΔF, %)		1.02	1.03	0.95	1.14	1.02	0.94
Maximum coefficient of inbreeding (%)		43.03	55.04	49.61	43.03	55.04	49.61
Inbred animals (%)		71.96	33.48	43.11	39.84	24.94	5.18
Highly inbred animals (%)		48.08	26.04	40.34	30.55	19.41	4.60
Average kinship or coancestry (C, %)		0.60	5.57	2.06	0.72	5.62	2.13
Average relatedness (ΔR, %)		1.21	11.13	4.12	1.42	11.25	4.25
Non-random mating rate (α)		0.06	0.03	0.01	0.08	0.03	0.01
Genetic Conservation index (GCI)		9.65	9.34	9.11	11.50	9.74	9.38

**Table 5 animals-11-00269-t005:** Measures of genetic variability and analysis of gene origin, effective number of non-founders (*N_ef_*), number of founder equivalents (*f_e_*), effective number of ancestors (*f_a_*) of Arabian purebred (PRá), Spanish purebred (PRE) and Hispano-Arabian (Há) horse breeds.

Parameter	PRá		PRE		Há	
	Historic	Current	Historic	Current	Historic	Current
Historical population	23,293	13,586	172,797	141,358	11,010	9997
Base population (one or more unknown parents)	1975	199	1110	33	406	362
Actual base population (one unknown parent = half founder)	257	68	483	15	19	15
Number of founders, n	1718	131	625	18	387	347
Number of ancestors, n	56	958	379	192	876	847
Effective number of non-founders (*N_ef_*)	14.73	10.00	12.06	11.40	24.99	23.57
Number of founder equivalents (*f_e_*)	43.76	37.67	20.63	21.49	29.23	34.11
Effective number of ancestors (*f_a_*)	22	16	14	14	22	23
Founder genome equivalents (*f_g_*)	11.02	7.90	7.61	7.45	13.47	13.94
*f_a_/*f_e_* ratio*	0.50	0.42	0.68	0.65	0.75	0.67
*f_g/_f_e_ ratio*	0.25	0.21	0.37	0.35	0.46	0.41

**Table 6 animals-11-00269-t006:** Summary of the results of Pillai’s Trace of Equality of Covariance Matrices of Canonical Discriminant Functions.

**Pillai’s Trace Criterion**	0.951
**F**	675,108.085
**Hypothesis df**	6
**Error df**	207,094
**Sig.**	0.001

**Table 7 animals-11-00269-t007:** Canonical Discriminant analysis efficiency parameters.

Test of Function(s)	Wilks’ Lambda	Chi-Square	df	Sig.
1 through 2	0.181	354,364.706	12	0.001
2	0.944	11,883.207	5	0.001

**Table 8 animals-11-00269-t008:** Multicollinearity analysis of genetic diversity parameters.

Parameters/Statistics	Tolerance (1 − R^2^)	VIF
Genetic Conservation Index	0.449	2.225
Coancestry, %	0.593	1.688
Non-random mating degree (α)	0.900	1.112
Number of Maximum Generations, n	0.362	2.761
Number of Complete Generations, n	0.353	2.835
Offspring number, n	0.988	1.012

Interpretation thumb rule: VIF = 1 (Not correlated); 1 < VIF < 5 (Moderately correlated); VIF ≥ 5 (Highly correlated).

**Table 9 animals-11-00269-t009:** Canonical variate pairs (discriminant functions) found in canonical discriminant analysis for genetic diversity parameters.

Function	1	2
Eigenvalue	4.227	0.059
Variance (proportion of discriminating ability), %	98.6	1.4
Canonical Correlation	0.899	0.236
Rc-Squared, Squared canonical correlation (shared variance), %	80.8	5.6

An efficient model will report a vale of >40% (0.4) for squared canonical correlations which translates into around 9% of explained variance among groups, provinces in our case.

**Table 10 animals-11-00269-t010:** Results for the tests of equality of group means to test for differences across breeds once redundant variables have been removed.

Rank of Variables	Wilk’s Lambda	df1	df2	df3	Exact F	df1	df2	Sig.	Rank
Offspring number, n	0.181	6	2	207,097	46,688.827	12	414,184	0.000	1
Average relatedness (AR), %	0.181	5	2	207,097	56,013.680	10	414,186	0.000	2
Non-random mating degree, α	0.190	4	2	207,097	67,054.671	8	414,188	0.000	3
Number of Maximum Generations, n	0.200	3	2	207,097	85,176.481	6	414,190	0.000	4
Number of Complete Generations, n	0.213	2	2	207,097	120,731.985	4	414,192	0.000	5
Genetic Conservation Index	0.235	1	2	207,097	337,293.642	2	207,097	0.000	6

F: Fisher-Snedecor approximation statistic; df1: numerator degrees of freedom; df2: denominator degrees of freedom, Rank denotes the importance of the discriminating power of a certain variable. As a rule of thumb, the closer Wilks’ lambda is to 0, the more the variable contributes to the discriminant function, hence placed at higher positions in the rank.

**Table 11 animals-11-00269-t011:** Standardized coefficients for zoometric variables.

Items	Function	
	F1	F2
Average relatedness (AR), %	1.083	0.071
Genetic Conservation Index (GCI)	−0.368	0.227
Non-random mating degree, α	−0.257	0.513
Number of Maximum Generations, n	0.475	−0.899
Number of Complete Generations, n	−0.380	0.887
Offspring number, n	0.008	0.062

Linear combination for a discriminant function (Z) could be described by F1 (Z) = µ1 Y1 + µ2 Y2 + ... + µi Yi, where µi is the canonical coefficient, and Yi are independent variables measured. F1 and F2: 1st and 2nd discriminant functions.

## Data Availability

The data presented in this study are available on request from the corresponding author.

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
