# Peer review of "Discriminant Canonical Analysis of the Contribution of Spanish and Arabian Purebred Horses to the Genetic Diversity and Population Structure of Hispano-Arabian Horses"

_animals, 2021, doi:10.3390/ani11020269_

Round 1

Reviewer 1 Report

The authors present an article on the genetic diversity and structure of horse breeds of Arabian, Spanish and Hispano-Arabian ancestry.

Overall, the article is of interest, there is a lot of work done, but I also feel there is a considerable lack of focus, and the authors should work now at improving how they communicate what is important here. I have also some methodological concerns and questions/suggestions:

Major comments:

- The general trend in Results is that values from Tables are just read, but a minimum interpretation on what these values mean - beyond the values already displayed in such Tables - is required. Some examples will be given below.

- CDA (1): A stepwise approach procedure is used to reduce the dimensions of the model. This approach is valid, but it comes with the problem that will completely remove the effect of the variable removed when they are (partially) correlated to another. I suggest using a regularized CDA here. It would be more meaningful on the parameters considered (centered and scaled), as it will retain all of them and inform on their relative weights.

- CDA (2): I have concerns with the sample sizes used, as they are highly imbalanced datasets (eg. in the historical pedigree, there are more than ten times more PRE individuals than Há). How is this accounted? In many classification algorithms is common practice to use sampling techniques (eg. upsampling, downsampling, SMOTE, ...) to account for this and avoid sources of bias.

- CDA validation: it is not completely clear to me how CDA results are validated, and I feel it is underexplained, specially compared to other methods related to CDA analysis. Since CDA is central in this paper, and cross-validation is what many readers will focus on concerning the credibility of the CDA results, please elaborate more on how the validation scheme was implemented (eg. there were train/test sets?), as well as its results.

- Many of the parameters estimated are actually based on inbreeding/coancestry and it is important to make sure these values are precise. Given that some generations/breeds have low pedigree completeness, how this impacts the estimates obtained for inbreeding/coancestry? to what extent the measure F is higher in PRE because of its higher overall pedigree completeness? and not because of its management/demographic history? What is the population size within these generations?

Minor comments:

- There are many parameters that are estimated - and its good to have them - but some are never discussed, are secondary, are redundant to others (eg. GD to F), or do not really provide an answer for the objectives in this work. The article needs to be more readable if it focus more on the parameters that are used in the Discussion, and move the remaining ones to a Supplemental Material.

- What is the degree of overlap between the historical and the current population? The current population represents yet alive individuals, but given that the historical one extends up to individuals born in 2019 this distinction current/historical is completely confusing, and again the Discussion does not help to understand why this differentiation matters.

- Section 2.1: the level of description of the historical and current pedigrees is highly imbalanced, and we know little about the current population until tables 1 and 2 are reached in Results. I'd advocate to include these tables where first cited. Following this point, I'd also remove sentences such as 230-232, since these do not really inform on methods, and readers already expect to see these values in Results.

- Figure 1 shows a dramatic reduction in the number of births, but never is said why. If the historical pedigree has some sort of record bias, shouldn't this dataset be truncated? (eg. if some breeds are updated more often than others, or some births are recorded with years of delay, explaining the pattern observed)

- Lines 373-377: How should readers interpret differences in "complete generations" and "equivalent generations"? this is never discussed.

- On generation numbers analyses. In Results, values are given for 1st to 5th generation. The 1st is the one with "known parents" and higher pedigree completeness. This confuses me. If founders (generation = 0) are 1st generation's parents ... is pedigree completeness dropping with time? I feel I am missinterpreting this.

- For effective population sizes. How should we interpret differences observed between NeF and NeC?

- How are "inbred" and "highly inbred" individuals defined?

- Lines 506-509: Explain this. I assume it comes from the ratios between effective numbers of founders/ancestors or genome equivalents, but it needs to be clear to all readers.

- Tables: They should be improved to make them more readable. Many have redundant parameters (e.g. Table 3 does not need SEM having SD and N), or values that can be simplified (e.g. in Table 2, why not give the male/female ratio as number). Format also needs to be consistent (breeds sometimes in columns others in rows).

- The use of acronyms is not consistent. e.g. PRE or PRA (sometimes PRÁ or PRá) are defined but rarely used; inbreeding (F) are defined more than once, etc.
- Line 27: GCI not defined here.
- Line 159: "." -> " and"
- Line 170: needs rephrasing?
- Line 172: Section names should be more concise.
- Line 209: Table 5 is cited before 3 and 4. Revise tables/figures citations.
- Line 213: Remove the inline comment (and of course review whether all relevant citations are correctly included or not).
- Line 221: I assume that ∆IBD in the denominator should be ∆F as in reference [25]?. [25] is probably the right citation here.
- Line 290: This should be section 2.5.4. Fix section numbers.
- Line 324: "is" -> "are"
- Line 330: remove "you"
- Line 471: "The" -> "the"
- Line 481: ";5.62" -> "; 5.62"
- Line 514: "y" -> "and "
- Line 702: ")" -> "))"

Author Response

Comments and Suggestions for Authors

The authors present an article on the genetic diversity and structure of horse breeds of Arabian, Spanish and Hispano-Arabian ancestry.

Overall, the article is of interest, there is a lot of work done, but I also feel there is a considerable lack of focus, and the authors should work now at improving how they communicate what is important here. I have also some methodological concerns and questions/suggestions:

Response: We thank the reviewer for his/her kind comment and will approach each particular comment or suggestion and relate them point-by-point.

Major comments:

- The general trend in Results is that values from Tables are just read, but a minimum interpretation on what these values mean - beyond the values already displayed in such Tables - is required. Some examples will be given below.

Response: Results were revised and interpretation for each parameter for which it was missing was provided.

- CDA (1): A stepwise approach procedure is used to reduce the dimensions of the model. This approach is valid, but it comes with the problem that will completely remove the effect of the variable removed when they are (partially) correlated to another. I suggest using a regularized CDA here. It would be more meaningful on the parameters considered (centered and scaled), as it will retain all of them and inform on their relative weights.

Response: We agree with the reviewer and clarified the reasons to choose our approach in the body text. The choice to perform a forward stepwise analysis was made considering the following alternatives. On the one hand, the first option considered was to perform a regularized canonical discriminant analysis. Regularization has been reported to improve the estimate of covariance matrices in situations where the number of predictors is larger than the number of data, as in such cases regularization may lead to an improvement of the efficiency of discriminant analysis. However, this was not our case as the nature of the variables considered may lead to the occurrence of considerable problems of multicollinearity. Such multicollinearity problems may derive from the fact that some of the variables initially considered, were computed including others (which were included too) among the terms in their formulas. As a result, even if models were simplified, removed variables may still be considered somehow.

Additionally, the present analysis faced the analyses of highly unequal sample sizes. To approach this compromising situation, we used the approximation proposed by Roemisch, et al. [1] of a regularized stepwise discriminant analysis. Unequal group sample sizes may affect the quality of classification, not axes. For these  reasons, what we did as suggested by Tai and Pan [2] and Roemisch, et al. [1] was to regularize priors based on group sizes and use them instead of equal priors using the compute from group sizes from the prior probability option in SPSS version 25.0 [3].

Furthermore, even if unequal sample sizes are acceptable as reported by Poulsen and French [4], some requirements must still be fulfilled. For instance, the sample size of the smallest group needs to exceed the number of predictor variables. As a “rule of thumb”, the smallest sample size should be at least 20 for each 4 or 5 predictors. The maximum number of independent variables is n - 2, where n is the sample size. Although such a low sample size may be valid, it is not encouraged, and generally it is best to have 4 or 5 times as many observations and independent variables for discriminant approaches to be efficient. The present study satisfies this condition by far

On the other hand, we considered the stepwise canonical discriminant analysis but needed to decide on either to perform backward or forward stepwise variable selection approaches. As a drawback, stepwise residual sum of squares will typically be above that for best subset, if there is correlation between the predictors considered. For this reason, we performed a multicollinearity analysis and correlated variables exceeding minimally acceptable levels were discarded. Contextually, when regressors (variables) are independent, that is are not correlated, the variables chosen by either forwards or backwards stepwise selection methods are exactly the same. 

Conclusively, forward stepwise selection method was chosen given it is rather effective (less time-consuming) than backward selection methods and provided it does not need n to be strictly greater than the number of variables as a requirement, which may make this approach valid for future research facing the same sample contexts.

- CDA (2): I have concerns with the sample sizes used, as they are highly imbalanced datasets (eg. in the historical pedigree, there are more than ten times more PRE individuals than Há). How is this accounted? In many classification algorithms is common practice to use sampling techniques (eg. upsampling, downsampling, SMOTE, ...) to account for this and avoid sources of bias.

Response: As reported by Poulsen and French [4], unequal sample sizes are acceptable. The sample size of the smallest group needs to exceed the number of predictor variables. As a “rule of thumb”, the smallest sample size should be at least 20 for a few (4 or 5) predictors. The maximum number of independent variables is n - 2, where n is the sample size. While this low sample size may work, it is not encouraged, and generally it is best to have 4 or 5 times as many observations and independent variables. The present study satisfies this requirement by far, hence the potential distorting effects derived from unequal group sample sizes comparison are avoided. Furthermore, in case we did not fulfil the requirement for outputs and conclusions to be valid, unequal group sample sizes may affect the quality of classification, not the axes. For this reasons, what we did as suggested by Tai and Pan [2] and Roemisch, et al. [1] was to set priors based on group sizes and use them instead of equal priors. We clarified this in the body text.

- CDA validation: it is not completely clear to me how CDA results are validated, and I feel it is underexplained, specially compared to other methods related to CDA analysis. Since CDA is central in this paper, and cross-validation is what many readers will focus on concerning the credibility of the CDA results, please elaborate more on how the validation scheme was implemented (eg. there were train/test sets?), as well as its results.

Response: The percentage of correctly classified cases is called the hit ratio [5]. To establish whether the percentage of correctly classified cases is enough as to consider discriminant functions issue valid results, as a form of significance, we used the leave-one-out cross-validation option. As reported by Schneider [6], the leave-one-out cross validation is K-fold cross validation taken to its logical extreme, with K equal to N, the number of data points in the set. That means that N separate times, the function approximator is trained on all the data except for one point and a prediction is made for that point. In this context, cross validated discriminant analysis should be at least 25% greater than that obtained by chance for classification accuracy to be considered sufficiently achieved.

The validity of cross-validation can be supported by Press's Q significance test of classification accuracy for original against predicted group memberships. In opposition to t-test for groups of equal size, in Press' Q statistic, groups can be of unequal size [7]. Press' Q statistic can be used to compare the discriminating power of a cross validated function to a model classifying individuals at random (50% of the cases correctly classified), as follows;

Press’ Q= [N-(nK)]2/N(K-1)

(1)

Where N is the number of individuals in the sample, n is the number of observations correctly classified, K is the number of groups.

Afterwards, the value of Press’ Q statistic should be compared to the critical value of 6.63 for χ2 with one degree of freedom at a significance of 0.01. Under this assumption, when Press’ Q exceeds the critical value of χ2=6.63, cross-validated classification can be regarded as significantly better than chance.

- Many of the parameters estimated are actually based on inbreeding/coancestry and it is important to make sure these values are precise. Given that some generations/breeds have low pedigree completeness, how this impacts the estimates obtained for inbreeding/coancestry? to what extent the measure F is higher in PRE because of its higher overall pedigree completeness? and not because of its management/demographic history? What is the population size within these generations?

Response: First of all, we must consider that, as stated in the paper, Hispano-Arabian breed is a composite breed which derives from the crossing between Spanish and Arabian Purebred horses. In this context, although Table 1 presents values for pedigree completeness indices for the Historical and current populations of Arabian, Spanish and Hispano-Arabian horses, separately, genetic diversity and structure parameters were computed considering all of the animals as a single population, given the ancestors of Hispano-Arabian horses are comprised in Spanish and Arabian horse populations. To clarify this, Figure 2 depicts the evolution of pedigree completeness, inbreeding, and population across maximum generations. Figure 2 shows that pedigree completeness describes the same trends in both current and Historical population, with a slight increase in the values for current population, which suggests unknown pedigree knowledge may be typical of ancestral individuals, and may stop being representative once those individuals disappear, as their descendants progressively increase pedigree knowledge after each generation. Furthermore, inbreeding and coancestry levels are maintained almost constant across generations until the 21st generation is reached, even in cases of generations comprising considerably higher numbers of individuals. These results suggest that the analysis of genetic diversity and population structure analysis is accurate and valid conclusions can be drawn.

Minor comments:

- There are many parameters that are estimated - and its good to have them - but some are never discussed, are secondary, are redundant to others (eg. GD to F), or do not really provide an answer for the objectives in this work. The article needs to be more readable if it focus more on the parameters that are used in the Discussion, and move the remaining ones to a Supplemental Material.

Response: Reviewer suggestion was followed and material which was not explicitly dealt with in discussion was moved to Supplementary Material.

- What is the degree of overlap between the historical and the current population? The current population represents yet alive individuals, but given that the historical one extends up to individuals born in 2019 this distinction current/historical is completely confusing, and again the Discussion does not help to understand why this differentiation matters.

Response: We totally understad the reviewer’s concern. We forgot to clarify, but we have added information in these regards so as to define each population set in a good manner. Differentiation matters as some parameters, such as generation intervals, require that currently alive populations are identified and computed separately.

- Section 2.1: the level of description of the historical and current pedigrees is highly imbalanced, and we know little about the current population until tables 1 and 2 are reached in Results. I'd advocate to include these tables where first cited. Following this point, I'd also remove sentences such as 230-232, since these do not really inform on methods, and readers already expect to see these values in Results.

Response: Reviewer suggestion was followed.

- Figure 1 shows a dramatic reduction in the number of births, but never is said why. If the historical pedigree has some sort of record bias, shouldn't this dataset be truncated? (eg. if some breeds are updated more often than others, or some births are recorded with years of delay, explaining the pattern observed)

Response: The drastic reduction in the number of Spanish Purebred horses was a direct consequence of the economic crisis whose effects in the equine sector became patent in 2008, when the real estate bubble burst [8]. According to Palomo [8], prior to 2008, Spanish Purebred horses had been acquired as a luxury item by real estate developers or companies, who had no choice but to missell them when the economic crisis arose. Abandonment, giveaway prices or slaughterhouse became the destination of thousands of individuals, which also brought about the drastic stop of breeding practices. It was only from 2012 on that it began to rebound due to leisure or sport. At the worst of times, individuals were sold for 150 euros, while the average price ranges between 3,000 and 5,000 euros. Hispano-Arabian horses did not suffer from the effects of the crisis as drastically as Spanish Purebred horses as these were usually kept on large farms, sometimes as a secondary production alongside cow farming; hence they were sold more easily and at a better price given their versatility [9].

- Lines 373-377: How should readers interpret differences in "complete generations" and "equivalent generations"? this is never discussed.

Response: As it can be observed, historical and current average number of equivalent generations, that is the number of generations separating the individual apart from each known ancestor tends to converge. This situation may be promoted as the animals presenting rather incomplete pedigrees are also the oldest ones, and have died in most of the cases, hence they are no longer considered to compute current population’s parameters, which only comprises alive individuals. However, the number of complete generations, or number of generations separating an individual from the furthest generation for which two of its ancestors are known is around half the number of equivalent generations, which could have been expected, but which suggests that even if the genealogical information of individuals is progressively increasing through the years, incomplete and partially incomplete pedigrees are still representative in the population from around the fourth to the fifth generation on, which can also be evidenced by the decrease in pedigree completeness occurring from the fourth generation on shown in Figure 2.

- On generation numbers analyses. In Results, values are given for 1st to 5th generation. The 1st is the one with "known parents" and higher pedigree completeness. This confuses me. If founders (generation = 0) are 1st generation's parents ... is pedigree completeness dropping with time? I feel I am missinterpreting this.

Response: There is only one misinterpretation. As it has been exhaustively reported in literature, pedigree genealogical known information (completeness) may progressively decrease in further generations from the first to the ith generation of ancestors. The misinterpretation lies in the fact that pedigree does not fall in time, indeed pedigree information normally tends to increase if populations are genealogically handled well.

Contextually, reduced values for pedigree completeness may derive from the fact that it is easier to know information about the parents of a certain animal than from its grandparents, great grandparents, and so on, as when older generations are considered, the knowledge about them is harder to control. This is extreme in the case of founding populations, for which pedigree completeness is 0, as not even information from parents is present. As I said, we did not specify this because is globally known.

- For effective population sizes. How should we interpret differences observed between NeF and NeC?

Response: As suggested by Gutiérrez, et al. [10], valuable information in regards population structure can be inferred from the comparison between individual increase in inbreeding and coancestry rate pathways to compute effective population sizes. This occurs as the two parameters are assumed to be measures of the same accumulated drift process, from the founding population to the present time. For instance, as reported by Malhado, et al. [11] both measurements of effective population size would be asymptotically equivalent in an idealized population and the disagreement between them may be  mainly caused by their differential ability to assess the effect of preferential matings. The imbalance between effective population size calculation via individual inbreeding and coancestry rate suggests the introduction of new reproductive individuals and the design of a plan for recommendable breeding pairs within the breeds populations so as to maintain the lowest possible ΔR coefficients, thereby preventing the increase in the probability of inbreeding deleterious effects from occurring and increasing the future genetic variability and effective size of the breeds.

- How are "inbred" and "highly inbred" individuals defined?

Response: Animals presenting any level of inbreeding different from 0 were considered to be inbred. However, as suggested by Beuchat [12], even if the deleterious effects of inbreeding begin to become evident at an inbreeding level of around 5%, it is when inbreeding reaches 10%, when there is significant loss of vitality in the offspring as well as an increase in the expression of deleterious recessive mutations. Hence, animals presenting values over 10% for inbreeding were considered to be highly inbred animals.

- Lines 506-509: Explain this. I assume it comes from the ratios between effective numbers of founders/ancestors or genome equivalents, but it needs to be clear to all readers.

Response: We clarified it.

- Tables: They should be improved to make them more readable. Many have redundant parameters (e.g. Table 3 does not need SEM having SD and N), or values that can be simplified (e.g. in Table 2, why not give the male/female ratio as number). Format also needs to be consistent (breeds sometimes in columns others in rows).

Response: Redundant parameters were removed and values suggested were simplified. Breed were placed in rows or columns to fit the format of tables so as for them to be presented in the most easily readable position as sometimes placing the breed at a different position may have cut numbers or titles. We also consulted the guide for authors and this is not a requirement.

- The use of acronyms is not consistent. e.g. PRE or PRA (sometimes PRÁ or PRá) are defined but rarely used; inbreeding (F) are defined more than once, etc.

Response: We checked across the manuscript to ensure the use of PRE for Spanish Purebred horses, PRá for Arabian horses and Há for Hispano-Arabian horses. We changed inbreeding by F the second time it was defined.

- Line 27: GCI not defined here.

Response: We defined it.

- Line 159: "." -> " and"

Response: Corrected.

- Line 170: needs rephrasing?

Response: We rephrased it.

- Line 172: Section names should be more concise.

Response: Long section names were reduced.

- Line 209: Table 5 is cited before 3 and 4. Revise tables/figures citations.

Response: table and Figure citations were revised and corrected.

- Line 213: Remove the inline comment (and of course review whether all relevant citations are correctly included or not).

Response: We removed it and we checked the introduction of references was correct.

- Line 221: I assume that ∆IBD in the denominator should be ∆F as in reference [25]?. [25] is probably the right citation here.

Response: Reference is correct here, IBD is identity by descent and it refers to the calculation of effective size via individual increase in inbreeding as reviewer suggested but also to the calculation of effective size via average coancestry as suggested below.

- Line 290: This should be section 2.5.4. Fix section numbers.

Response: Corrected.

- Line 324: "is" -> "are"

Response: Corrected.

- Line 330: remove "you"

Response: Corrected.

- Line 471: "The" -> "the"

Response: Corrected.

- Line 481: ";5.62" -> "; 5.62"

Response: Corrected.

- Line 514: "y" -> "and "

Response: Corrected.

- Line 702: ")" -> "))"

Response: Corrected.

Reviewer 2 Report

General comment:

  • I am positively impressed by the amount of work done in the manuscript. However, unfortunately, the manuscript in the present form is very long and difficult to follow. I think it could be shortened by excluding some redundant information in the introduction and results and some extensive details in the M&M.

Another possible solution is to split the manuscript in two papers:

  1. To explore genetic diversity in the three breeds: Arabian, Hispano-Arabian, Spanish
  2. Use the discriminant analysis to evaluate how genetic parameters are able to discriminate those populations and evaluate distance/vicinity among them

  • An English review is needed. Some typos are present and sometime the sentences are long and difficult to follow.
  • I would replace “conforming breed” with a more appropriate term.. such as: ancestor breed throughout all the manuscript
  • in all the manuscript use the “,” to separate thousands

Specific comments: 

Abstract:

L36 here you present extending the generation interval to reduce inbreeding, however you did not mention the level of inbreeding in your population. So please, either remove this sentence or provide the info about average inbreeding and inbreeding trend.

L39 – L43 similarly since you did not provide information in the abstract about the level of genetic diversity loss in your populations, those thoughts are difficult to follow.  

Introduction:

L62 – L64 suggest rephrasing as follow: even though a small number of pure Arabian horses were brought by the army of Baly, they were sufficient to originate a new type of horse: the Iberian Arabian horse

L65 please replace “giving way to”  to “created” or “founded”

L74 Please specify better what you mean with conforming breeds

L74-76 more closely related to each other? Please rephrase because a bit unclear

L87 suggestion to rephrase: However, the formal creation of the breed dates to the 1883, where an official breeding program was implemented together with the introduction of Arabian horses to upgrade and expand other horse breeding programs.

From L89 to L93 this sentence is very long, please rephrase

L103 rephrase without using future tense: e.g.: As a result, in the middle of the 19th century the breed has started to be considered standardized, while being frequently used in the military campaigns.

L108 – 110 I think this sentence is not really needed as those concepts were already mentioned above

L114 “marked” without capital letter

L114 can you specify how many foundation stallions?

L114 I would suggest specifying already in the text (not within parenthesis) what you mean with 50% e.g., 50% Arabian and 50% purebred

L122 remove “in one their herds” since it does not provide any specific information

L122 “which were not been frequently used” instead of “had not frequently been performed”

L124 what do you mean with “quality” F2 and F3?

L125 “matured”?

L126 I think “accounting” here can be removed

L144 replace flux with flow

Material and methods

L154 remove the “-“ before 4268

L155 replace “provided “ with Since … is the product of the cross …

L159 remove dot after (ANCCE)

L170 and on the currently..

2.2 suggest making a more condense version of the title

L179 should it be:  ?

replace “pf” with of

L232 why are you referring here to a result table? I think this sentence should be remove or paste into the result section

L258 – 265 this part is very long and not really needed in the material and methods section. I think it would be better to just explicitly say which value of VIF you used and then perhaps use that information as part of the discussion.

L284 to L289: From the number you reported above I do not see the issue of small sample size.. Why do you report that information here?

L291 to L297 Since the manuscript is very long, I would skip that information and start directly with the method you used (Pillai)

L308 if the PCA was computed as preliminary analysis and before CDA I would put this paragraph before the CDA paragraph.

L310 – 311 this is a result

L343 – L344 Is the Nei’s genetic distance rightly placed here?

Results:

L411 – L414 sorry but I do not understand how you presented the results of the portion of stallion per mares. Which figure do you want to present here? The percentage of male and female in your population or on average how many females per stallion??

L418 – L421 this result contradicts what previously state at L408

L424 – L427 this seems a repetition of L414 to L418. Please read carefully your manuscript and avoid repeating information. It makes everything much difficult to follow and not very pleasant to read.

L429-L434 Are really so old and still used for breeding??

Table 4 maybe as supplementary?

L454 – L457 I think it is already clear from the above sentence, thus I would remove this explanation as it is redundant.

L464 onwards: the explanation of the table should come first and after the table (in this case table 5 but also the same is valid for other tables such as table 8, 9)

L464 How do you define highly inbred animals?

L515 unclear who are those single ancestor... why did you choose them?

Discussion:

In general, interesting discussion.

L716 Sorry but I do not understand the connection of the sentence above and your results. If I understood correctly, you used pedigree data dating back at latest to 1900… how can your results contribute to explain somethings happened in 742? Or even in 1609?

Author Response

Reviewer 2

  • I am positively impressed by the amount of work done in the manuscript. However, unfortunately, the manuscript in the present form is very long and difficult to follow. I think it could be shortened by excluding some redundant information in the introduction and results and some extensive details in the M&M.

Response: We would like to thank the reviewer for his/her comments and will try to address them as appropriately as possible in order to improve readability and manuscript flow.

Another possible solution is to split the manuscript in two papers:

  1. To explore genetic diversity in the three breeds: Arabian, Hispano-Arabian, Spanish
  2. Use the discriminant analysis to evaluate how genetic parameters are able to discriminate those populations and evaluate distance/vicinity among them

Response: Although we considered this option at first, we reviewed literature and determine that the mere exploration of genetic diversity may not be a completely novel approach. Hence, we decided to work on the present paper.

  • An English review is needed. Some typos are present and sometime the sentences are long and difficult to follow.
  •  

Response: English language was revised and checked by a Cambridge University ESOL examination instructor to detect and correct typos and grammar mistakes and to improve overall readability.

  • I would replace “conforming breed” with a more appropriate term.. such as: ancestor breed throughout all the manuscript

Response: The term was replaced as suggested.

  • in all the manuscript use the “,” to separate thousands

Response: “,” was used to separate thousands along the manuscript.

Specific comments: 

Abstract:

L36 here you present extending the generation interval to reduce inbreeding, however you did not mention the level of inbreeding in your population. So please, either remove this sentence or provide the info about average inbreeding and inbreeding trend.

Response: Information requested was added.

L39 – L43 similarly since you did not provide information in the abstract about the level of genetic diversity loss in your populations, those thoughts are difficult to follow.  

Response: Current diversity loss levels were added to the abstract as suggested.

Introduction:

L62 – L64 suggest rephrasing as follow: even though a small number of pure Arabian horses were brought by the army of Baly, they were sufficient to originate a new type of horse: the Iberian Arabian horse

Response: Rephrased as suggested.

L65 please replace “giving way to”  to “created” or “founded”

Response: Changed to created.

L74 Please specify better what you mean with conforming breeds

Response: We replaced the term conforming breed to ancestor breed as suggested by reviewer as we fell it expresses better the idea that we want to transmit of two breeds being crossed to develop a composite breed.

L74-76 more closely related to each other? Please rephrase because a bit nuclear

Response: Rephrased as suggested.

L87 suggestion to rephrase: However, the formal creation of the breed dates to the 1883, where an official breeding program was implemented together with the introduction of Arabian horses to upgrade and expand other horse breeding programs.

Response: Rephrased as suggested.

From L89 to L93 this sentence is very long, please rephrase

Response: Sentence was divided in to two shorter sentences.

L103 rephrase without using future tense: e.g.: As a result, in the middle of the 19th century the breed has started to be considered standardized, while being frequently used in the military campaigns.

Response: We rephrased the sentence as suggested.

L108 – 110 I think this sentence is not really needed as those concepts were already mentioned above

Response: Removed.

L114 “marked” without capital letter

Response: Corrected.

L114 can you specify how many foundation stallions?

Response: Yes, 16

L114 I would suggest specifying already in the text (not within parenthesis) what you mean with 50% e.g., 50% Arabian and 50% purebred

Response: We specified.

L122 remove “in one their herds” since it does not provide any specific information

Response: Removed.

L122 “which were not been frequently used” instead of “had not frequently been performed”

Response: Changed.

L124 what do you mean with “quality” F2 and F3?

Response: We clarified. Optimally fitting the standard of the breed.

L125 “matured”?

Response: We meant adult, but it is unnecesary, we removed it.

L126 I think “accounting” here can be removed

Response: Removed.

L144 replace flux with flow

Response: Changed.

Material and methods

L154 remove the “-“ before 4268

Response: Removed.

L155 replace “provided “ with Since … is the product of the cross …

Response: Changed.

L159 remove dot after (ANCCE)

Response: Removed.

L170 and on the currently..

Response: Changed.

2.2 suggest making a more condense version of the title

Response: Title was condensed.

L179 should it be:  ?

Response: Changed.

replace “pf” with of

Response: Changed.

L232 why are you referring here to a result table? I think this sentence should be remove or paste into the result section

Response: Sentence was removed as suggested.

L258 – 265 this part is very long and not really needed in the material and methods section. I think it would be better to just explicitly say which value of VIF you used and then perhaps use that information as part of the discussion.

Response: We removed it.

L284 to L289: From the number you reported above I do not see the issue of small sample size.. Why do you report that information here?

Response: We removed it.

L291 to L297 Since the manuscript is very long, I would skip that information and start directly with the method you used (Pillai)

Response: We removed it.

L308 if the PCA was computed as preliminary analysis and before CDA I would put this paragraph before the CDA paragraph.

Response: Suggestion was followed.

L310 – 311 this is a result

Response: We changed to results section.

L343 – L344 Is the Nei’s genetic distance rightly placed here?

 Response: Yes, we justified its inclusion here.

Results:

L411 – L414 sorry but I do not understand how you presented the results of the portion of stallion per mares. Which figure do you want to present here? The percentage of male and female in your population or on average how many females per stallion??

Response: There as a mistake we corrected it, it was how many stallions are there in the population per female for each of the three breeds.

L418 – L421 this result contradicts what previously state at L408

Response: It does not, it rather complements it as what is being reported here is the total percentage of the total offspring that is actually selected for breeding.

L424 – L427 this seems a repetition of L414 to L418. Please read carefully your manuscript and avoid repeating information. It makes everything much difficult to follow and not very pleasant to read.

Response: We apologize. We removed it.

L429-L434 Are really so old and still used for breeding??

Response: Numbers reflect the information in the databases that were provided to us. Some stallions and mares may be used as breeding animals after they have deceased (using artificial insemination or embryo transfer), hence, increasing the age at which such animals are used for breeding. Furthermore, the high values for SD are indicative of the presence of a wide range of variation for this parameter, which suggests many individuals are indeed bred at a much lower  age than the average.

Table 4 maybe as supplementary?

Response: We moved Table 4 and others to supplementary material as suggested by other reviewers.

L454 – L457 I think it is already clear from the above sentence, thus I would remove this explanation as it is redundant.

Response: We removed it.

L464 onwards: the explanation of the table should come first and after the table (in this case table 5 but also the same is valid for other tables such as table 8, 9)

Response: We followed reviewer suggestion.

L464 How do you define highly inbred animals?

Response: Animals presenting any level of inbreeding different from 0 were considered to be inbred. However, as suggested by Beuchat [12], even if the deleterious effects of inbreeding begin to become evident at an inbreeding level of around 5%, it is when inbreeding reaches 10%, when there is significant loss of vitality in the offspring as well as an increase in the expression of deleterious recessive mutations. Hence, animals presenting values over 10% for inbreeding were considered to be highly inbred animals.

L515 unclear who are those single ancestor... why did you choose them?

Response: We did not choose them. Endog software identifies and ranks the individuals considering their marginal contribution to diversity. We just determined which was the one accounting for the highest marginal contribution to diversity of the whole the population.

Discussion:

In general, interesting discussion.

Response: We thank the reviewer for his/her kind comment.

L716 Sorry but I do not understand the connection of the sentence above and your results. If I understood correctly, you used pedigree data dating back at latest to 1900… how can your results contribute to explain somethings happened in 742? Or even in 1609?

Response: We understand and agree with reviewer comment. What we wanted to express is that the historical context and the higher contribution of PRá to Há genetic diversity evidenced in the present study, even if the breeding practices seeking the obtention of Há horses would only become more relevant from 1800 on, may suggest the greater contribution of PRás to the development of Há breed has been patent from the very first crosses carried out after the first PRá horses came to the Iberian Peninsula in 742 and still continues in our times.

Reviewer 3 Report

In a time when more and more studies of equine diversity are based on genomic SNP analysis it is good to see a robust study based on pedigree analysis.

Other authors have made the case that where pedigrees are robust and complet ( >10 generations) there is good correlation between inbreeding assessed through pedigree analysis and runs of homozygosity ( Todd et al 2018 https://www.ncbi.nlm.nih.gov/pmc/articles/PMC5906619/ and Mancini et al 2020 https://www.mdpi.com/2076-2615/10/8/1310 )

I think reference to this correlation would help validate your research.

My principal concerns with your paper  are the need for some changes to tidy up use of English language. I have attached a marked up copy with some suggested revisions.

Whilst the focus of your paper is on the novel Canonical analysis I think that focus has taken your eye off of some findings from your data that are of significance and that you skim over.

In particular the levels of effective population size that you correctly assess from rate of change of inbreeding and not as so many authors erroneously do from Census and number of breeding males and females.

Table 8 shows Effective Population Size  of the Arabian Purebred population at 4.53% declining from 49.02% in the historic population. That rings major alarm bells and warrant comment!!!

In fact the Spanish Purebred and Hispano Arabian populations also have effective population sizes close to or below 50 which is the threshold below which the FAO of the United Nations recognise that populations may no longer be viable.

Given that you have the data to hand I would suggest running it on the PopRep servers at https://popreport.fli.de/cgi-bin/entry.pl

The inbreeding and population reports you get back from there would help corroborate the results you have got using Endog 4.8

(I declare an interest in having helped with the English translation of the Endog handbook!)

The monitoring report also has a decision making three which will help you decide which method of assessing Effective Population Size is most appropriate for your data

Your Figure 1  shows the population demographics of the 3 breeds 1900 to 2019.

There is a glaring drop off in the number of Spanish purebred foals being born in 2006. You offer no explanation and it is the most dramatic  component of Figure 1. Is it a registry error? a fault with the dataset? Or was there a real life drop in registration numbers? If so why?

Author Response

Reviewer 3

Comments and Suggestions for Authors

In a time when more and more studies of equine diversity are based on genomic SNP analysis it is good to see a robust study based on pedigree analysis.

Other authors have made the case that where pedigrees are robust and complet ( >10 generations) there is good correlation between inbreeding assessed through pedigree analysis and runs of homozygosity ( Todd et al 2018 https://www.ncbi.nlm.nih.gov/pmc/articles/PMC5906619/ and Mancini et al 2020 https://www.mdpi.com/2076-2615/10/8/1310 )

I think reference to this correlation would help validate your research.

Response: We really thank you for providing us with the references that the reviewer enclosed. We have found them very interesting and had read them to consider their information in the discussion of our paper. For sure, these references will be helpful for future papers as well.

My principal concerns with your paper  are the need for some changes to tidy up use of English language. I have attached a marked up copy with some suggested revisions.

Response: English language was revised and checked by a Cambridge University ESOL examination instructor to detect and correct typos and grammar mistakes and to improve overall readability.

Whilst the focus of your paper is on the novel Canonical analysis I think that focus has taken your eye off of some findings from your data that are of significance and that you skim over.

Response: We agree with the reviewer, but wanted to add that commenting each result would have made the paper too extense. We will approach each particular comment or suggestion and relate them point-by-point.

In particular the levels of effective population size that you correctly assess from rate of change of inbreeding and not as so many authors erroneously do from Census and number of breeding males and females.

Table 8 shows Effective Population Size of the Arabian Purebred population at 4.53% declining from 49.02% in the historic population. That rings major alarm bells and warrant comment!!!

Response: We have found out that Table was incorrect. This in fact was a typo which has been corrected. Current effective population via individual increase in inbreeding (NeFi) is indeed 48.54, not 4.53. Anyway, we commented the results.

In fact the Spanish Purebred and Hispano Arabian populations also have effective population sizes close to or below 50 which is the threshold below which the FAO of the United Nations recognise that populations may no longer be viable.

Response: In our opinion, this can be linked to the recent sharp bottleneck occurring due to the crisis, as afterwards, there may have been an overrepresentation of popular individuals in the populations considered in this study.

FAO/UNEP [13] proposed the general rule of maintaining rate of inbreeding per generation should not exceed 1-3% [14]. Higher rates fix deleterious recessive genes too rapidly for selection to eliminate them, and the vigour and fertility of the populations decrease. When inbreeding rate is below 1% populations have been partially purged of deleterious genes and tolerate higher rates of inbreeding, hence, animal breeders can safely ignore some inbreeding and random loss of genes. In endangered populations conservationists develop a rather conservative approach. The rate of loss per generation of heterozygosity due to inbreeding as measured by F is equal to 1/(2 Ne), where Ne is the effective population size.

The definition of Ne is complex, but certain criterion must be considered to permit a correct interpretation of this parameter. For instance, sex ratio must be equal and individuals must randomly mate. A number of additional “ideal” characteristics could be stated. In practice Ne is always smaller than the actual number of breeding individuals. Thus, Ne must equal at least 50 if our aim is to keep inbreeding rate below 1%. Still, even if inbreeding rate is 1%, the loss of genetic diversity is appreciable after a few generations, and a gradual erosion of genetic variation cannot be avoided. Eventually, the population will become virtually homozygous, the time depending on Ne. Consequently, 1% criterion must be viewed as short-term criterion. A population with an effective size of 50, will lose about 1/4 of its genetic diversity after 20 to 30 generations, and alongside with it, much of its capacity to adapt to changing conditions [15].

To maintain a genetically healthy population in these situations, Ne must be increased. FAO/UNEP [13] suggests that G must approximately equal to Ne, G being the number of generations the population is likely to retain its fitness at a relatively high level. Still, to conserve short-term fitness, or to maintain short-term fitness in captive populations other criteria must be accounted for, given effective population size is considerably affected by unbalanced sex ratios, population size evolution, by a non-random distribution of progeny among families, and other characteristics of the breeding systems implemented.

In line with these suggestions, Leroy, et al. [16] reported that restricting the number of generations when calculating effective size may be a good option for population monitoring, due to its effectivity to detect short-term changes in genetic diversity while it permits a generation scaled increased accuracy of the estimation of effective size, while reducing the bias related to ancestral pathways disequilibrium in pedigrees. Still, the consideration of very limited number of generations may not completely account for the lack of independence of family sizes across generations. As a result, for breeds with relatively complete pedigree records, as in our study, the estimation of effective size via coancestry rate may be of interest to provide an evaluation of changes in genetic diversity over long periods. 

Given that you have the data to hand I would suggest running it on the PopRep servers at https://popreport.fli.de/cgi-bin/entry.pl

The inbreeding and population reports you get back from there would help corroborate the results you have got using Endog 4.8

(I declare an interest in having helped with the English translation of the Endog handbook!)

The monitoring report also has a decision making three which will help you decide which method of assessing Effective Population Size is most appropriate for your data

Response: We performed the same analyses in PopRep servers as suggested and using CFC software by Sargolzaei, et al. [17] and results were comparable. Still, we spent a long time working on the preparation and curation of the database used in this study before Endog could even read it, hence, these results could have been expected somehow.

Your Figure 1  shows the population demographics of the 3 breeds 1900 to 2019.

There is a glaring drop off in the number of Spanish purebred foals being born in 2006. You offer no explanation and it is the most dramatic  component of Figure 1. Is it a registry error? a fault with the dataset? Or was there a real life drop in registration numbers? If so why?

 Response: The drastic reduction in the number of Spanish Purebred horses was a direct consequence of the economic crisis whose effects in the equine sector became patent in 2008, when the real estate bubble burst [8]. According to Palomo [8], prior to 2008, Spanish Purebred horses had been acquired as a luxury item by real estate developers or companies, who had no choice but to missell them when the economic crisis arose. Abandonment, giveaway prices or slaughterhouse became the destination of thousands of individuals, which also brought about the drastic stop of breeding practices. It was only from 2012 on that it began to rebound due to leisure or sport. At the worst of times, individuals were sold for 150 euros, while the average price ranges between 3,000 and 5,000 euros. Hispano-Arabian horses did not suffer from the effects of the crisis as drastically as Spanish Purebred horses as these were usually kept on large farms, sometimes as a secondary production alongside cow farming; hence they were sold more easily and at a better price given their versatility [9].

Round 2

Reviewer 1 Report

The authors responded all my comments, and the new version of the manuscript addresses all my previous concerns. The article also feels much more clear and focused now.

Minor comments:

- Figure 2: Labels, legend and figure foot could be improved, so it is easier to understand what is plotted (eg. it is not completely obvious that "Historic Population" and "Current Population" legend labels refer to Pedigree completeness).
- line 324: "need n to" -> "need to"
- line 412: remove trailing " V"
- line 516: "shown" -> "shown in"
- line 521: "population" -> "population size"
- 552: mind break line
- lines 867-869: mind bold text.
- Tables: not sure all supplementary tables are cited.

Author Response

Reviewer 1

Comments and Suggestions for Authors

The authors responded all my comments, and the new version of the manuscript addresses all my previous concerns. The article also feels much more clear and focused now.

Response: We thank the reviewer for his/her kind comments.

Minor comments:

- Figure 2: Labels, legend and figure foot could be improved, so it is easier to understand what is plotted (eg. it is not completely obvious that "Historic Population" and "Current Population" legend labels refer to Pedigree completeness).

Response: We clarified legends as suggested by the reviewer.

- line 324: "need n to" -> "need to"

Response: Changed.

- line 412: remove trailing " V"

Response: Removed.

- line 516: "shown" -> "shown in"

Response: Added.

- line 521: "population" -> "population size"

Response: Changed.

- 552: mind break line

Response: Corrected.

- lines 867-869: mind bold text.

Response: Bold text was turned to regular fonts.

- Tables: not sure all supplementary tables are cited.

Response: We revised and corrected Supplementary Tables citation.

Reviewer 2 Report

General comment:

The manuscript has improved massively, and I think it has a good scientific soundness. However, a thorough English review must be done before submitting again since the readability is still weak.

L38: please rephrase as follow: Pedigree records were available for 207,100 animals born between 1884 and 2019.

L41: please rephrase to: Increase the length of the generation interval might be an effective solution to reduce the increase in inbreeding found in the studied breeds (.....)

L88-L90 this sentence is not clear

L106 “the popularity the cross of Há horses with thoroughbred part-bred, to conform the Tres Sangres (Three bloods) composite breed” does not make a lot of sense. Please rephrase.

L112 in this sentence there is two times the subject of the predicate… its or  the Hispano-ArabianHá horse breed… both in the same sentence are a repetition.

L136 “five new generationS”

L153 previous analyses instead of “analyses carried out” did not consider instead of never considered

L171 remove “which were considered in the pedigree analyses” as redundant information

L244-L250 since the manuscript is still very long, I would suggest to remove those lines as they are not really describing any methods but are just general considerations.

L292 to L326 would it be more appropriate as discussion?

L349 – 350 “The number of variables in the smallest sets  The maximum number of canonical correlations between two sets of variables “??? what does it mean? Please rephrase

L475 to L 484 is discussion of the results

L481 I think this sentence is more appropriate after the sentence finishing with: which also brought about the drastic stop of breeding practices.

Figure 2 the decimal for the pedigree completeness must be separated by “.”.

L532-534 discussion

L867 – 869 is in bold

L923 advantage to meet, remove of

Author Response

Reviewer 2

Comments and Suggestions for Authors

General comment:

The manuscript has improved massively, and I think it has a good scientific soundness. However, a thorough English review must be done before submitting again since the readability is still weak.

Response: We thank the reviewer for his/her kind comments and we revised the manuscript again to improve readability.

L38: please rephrase as follow: Pedigree records were available for 207,100 animals born between 1884 and 2019.

Response: Rephrased.

L41: please rephrase to: Increase the length of the generation interval might be an effective solution to reduce the increase in inbreeding found in the studied breeds (.....)

Response: Rephrased.

L88-L90 this sentence is not clear

Response: Clarified.

L106 “the popularity the cross of Há horses with thoroughbred part-bred, to conform the Tres Sangres (Three bloods) composite breed” does not make a lot of sense. Please rephrase.

Response: Sentence was rephrased.

L112 in this sentence there is two times the subject of the predicate… its or  the Hispano-ArabianHá horse breed… both in the same sentence are a repetition.

Response: Corrected.

Response:

L136 “five new generationS”

Response: Corrected.

L153 previous analyses instead of “analyses carried out” did not consider instead of never considered

Response: Corrected.

L171 remove “which were considered in the pedigree analyses” as redundant information

Response: Removed.

L244-L250 since the manuscript is still very long, I would suggest to remove those lines as they are not really describing any methods but are just general considerations.

Response: Removed.

L292 to L326 would it be more appropriate as discussion?

Response: We feel this should be kept here as this fragment does not discuss any of the results, but the reasons why we decided to perform the analyses that were developed in the manuscript. Hence it is rather methods.

L349 – 350 “The number of variables in the smallest sets  The maximum number of canonical correlations between two sets of variables “??? what does it mean? Please rephrase

Response: Sentence was rephrased.

L475 to L 484 is discussion of the results

 Response: We changed this ffragment to discussion as suggested.

L481 I think this sentence is more appropriate after the sentence finishing with: which also brought about the drastic stop of breeding practices.

Response: Moved.

Figure 2 the decimal for the pedigree completeness must be separated by “.”.

Response: We corrected it.

L532-534 discussion

Response: Moved.

L867 – 869 is in bold

Response: Corrected.

L923 advantage to meet, remove of

Response: Removed.